# FreeEyeglass: Training-free and Target-mask-free Eyeglass Transfer for Facial Videos

**Weng Ian Chan**                                             *chan.wengian@ist.osaka-u.ac.jp*
*The University of Osaka*

**Yuantian Huang**                                      *huang_yuantian@cyberagent.co.jp*
*CyberAgent Inc.*

**Xingchao Yang**                                           *you_koutyo@cyberagent.co.jp*
*CyberAgent Inc.*

**Fumio Okura**                                                   *okura@ist.osaka-u.ac.jp*
*The University of Osaka*

**Takafumi Taketomi**                                 *taketomi_takafumi@cyberagent.co.jp*
*CyberAgent Inc.*

**Reviewed on OpenReview:** *https://openreview.net/forum?id=6aFRoQcm3H*

## Abstract

The rise of e-commerce and short-video platforms has fueled demand for realistic video-based virtual try-on. Unlike virtual try-on of clothing, which has been actively studied to date, virtual try-on of eyeglasses is uniquely challenging: they align closely with facial structure and strongly affect facial identity, making the faithful preservation of unedited regions especially important. Existing generative editing approaches, such as GAN- and diffusion-based methods, lack reconstruction objectives and often rely on inpainting, which fails to ensure identity consistency. We argue that semantic editing requires not only plausible generation but also faithful reconstruction, making autoencoder-based latent spaces a natural fit. We introduce a training-free, reference-guided framework for video eyeglass transfer built on Diffusion Autoencoders (DiffAE). By blending semantic features in the encoder and incorporating spatial-temporal self-attention, our method achieves realistic, identity-preserving, and temporally consistent results, and points to the potential of autoencoder-based latent spaces for local video editing. The project page is available at `https://moegi161.github.io/freeeyeglass-project/`.

## 1 Introduction

The rapid growth of e-commerce and short-video platforms has created a strong demand for realistic video-based virtual try-on systems. While most existing research has centered on clothing in the video setting (Jiang et al., 2022; Xu et al., 2024; Fang et al., 2024; Wang et al., 2024b; Nguyen et al., 2025), work on other wearable objects has been explored only in the image domain (Miao et al., 2025; Feng et al., 2025). Among them, eyeglasses stand out as a particularly important category. They have long been treated as a standalone research topic in vision and graphics, with prior work on try-on (Zhang et al., 2017; Li et al., 2023), detection (Wu et al., 2002; Bekhet & Alahmer, 2021), removal (Lyu et al., 2022; Zhang & Guo, 2025), and product design (Bai et al., 2021; Plesh et al., 2023). Compared to clothing and other accessories, eyeglasses pose uniquely complex challenges. Geometrically, they must fit precisely on the nose and ears rather than being simply overlaid on the face. They also overlap directly with the eyes and eyebrows, regions known to be critical for facial identity perception (Schyns et al., 2002; Tanaka & Simonyi, 2016). Realistic video

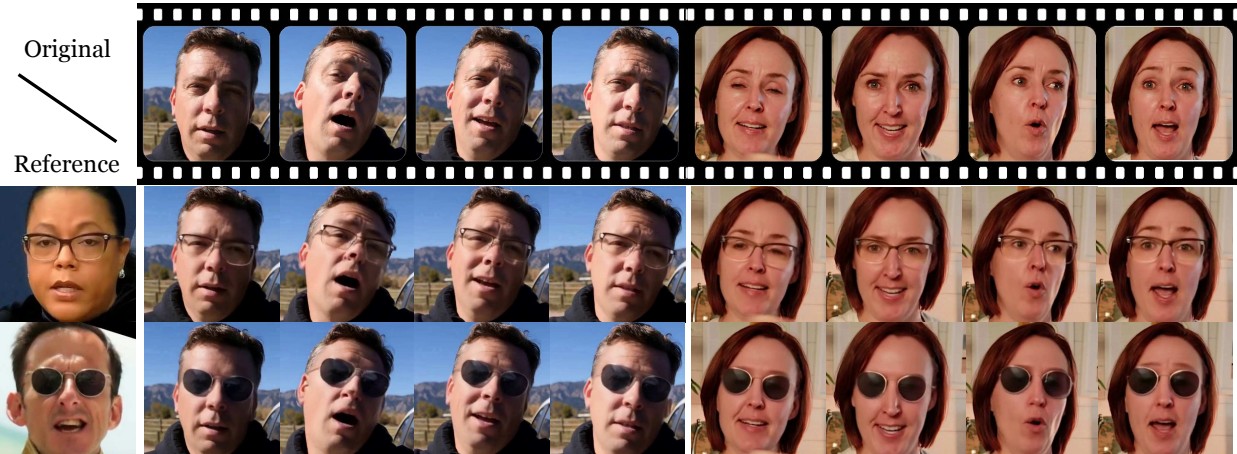

Figure 1: **FreeEyeglass** is a training-free reference-based video editing method for transferring reference eyeglasses to target facial videos semantically while achieving harmonious local editing and temporal consistency. Given a reference facial image specified with eyeglass position, our method does not require any frame-by-frame accurate masks on the target videos for transferring eyeglasses, which has been required in state-of-the-art inpainting-based editing methods.

eyeglass transfer, therefore, requires not only placing the glasses plausibly but also preserving identity and maintaining temporal coherence under motion and viewpoint changes. Despite this practical importance, the problem has not been systematically studied to date.

Despite rapid advances in semantic image and video editing, the generative models that are widely adopted, GANs (Karras et al., 2019; 2020b;a) and text-conditional diffusion models (Rombach et al., 2022), are not inherently well-suited for local editing tasks such as object transfer. Trained primarily for generation, these models typically lack reconstruction ability, making them prone to altering unedited regions when adapted to editing. To enforce more localized changes, many object transfer methods (Yang et al., 2023a; Chen et al., 2024b;a; Song et al., 2024; Jiang et al., 2025) train or finetune these models with an inpainting objective, where masked regions in target images are filled with reference content. While this improves locality, they still cannot handle the unique challenges of eyeglass transfer. Target masks inevitably discard identity-related content and spatial information around the eyes, leading to artifacts, poor harmonization, and identity inconsistency. These issues are further amplified in videos, where temporal coherence must also be maintained, and the results often resemble simply copying the reference into the target region.

These shortcomings indicate that eyeglass transfer requires more than inpainting-based generation. What is needed is an approach that can edit locally while preserving critical identity information. Generative models that lack reconstruction objectives struggle to provide this guarantee. Autoencoders, in contrast, are trained with explicit reconstruction objectives, equipping them with a strong capability to retain unedited content. Diffusion autoencoders (DiffAE) (Preechakul et al., 2022) further show that their latent spaces are compact and semantically structured, enabling natural local edits with minimal distortion. Through extensive experimentation, we observe that DiffAE's semantic latent space preserves appearance while implicitly adapting the placement and structure of eyeglasses to the target face. Even with a coarse, face-aligned reference mask and without explicit geometric modeling, the transferred eyeglasses align well with the target across frames. This observation suggests that DiffAE's semantic latent space, despite not being trained for any generative editing objectives, provides a practical and effective solution for eyeglass transfer in video, where both identity preservation and local realism are critical.

Building on this insight, we leverage DiffAE as a backbone and introduce a training-free, target-mask-free framework for reference-based eyeglass transfer in video, as shown in Fig. 1. While DiffAE has been explored with text or classifier guidance (Kim et al., 2023), the approach to achieving reference-based editing with this model has not been studied. We address this by designing a feature-blending mechanism in the semantic

encoder that combines reference and target features to construct new semantic features for each frame. These features are used in DDIM inversion with noise blending to guide the placing of eyeglasses on the target face. Furthermore, we extend this framework to video by incorporating spatial-temporal self-attention focused on the editing region to enhance temporal consistency. Our method achieves realistic eyeglass transfer that preserves identity and adapts to pose changes. It demonstrates the potential of compact autoencoder latents for local video editing.

Our contributions are summarized as follows:

- We present the first *training-free* framework that requires *no target-video masks* for **reference-based video eyeglass transfer** to tackle a practically important yet underexplored problem in virtual try-on.
- We show how to adapt diffusion autoencoders for reference-based editing by designing a simple feature-blending strategy in the semantic encoder, combined with spatial-temporal self-attention to ensure natural placement and temporal consistency of eyeglasses.
- We establish a comprehensive benchmark for the video eyeglass transfer task, which will be publicly released (except for the CG face dataset).

## 2 Related Work

**Eyeglasses** Eyeglasses are a distinctive accessory that strongly influences facial perception and identity, and have therefore become a standalone research topic in computer vision and graphics. Prior works span several directions, including virtual try-on (Li & Yang, 2011; Yuan et al., 2011; Niswar et al., 2011; Huang et al., 2012; 2013; Tang et al., 2014; Zhang et al., 2017; Li et al., 2023), eyeglass detection (Wu et al., 2002; Bekhet & Alahmer, 2021), removal (Hu et al., 2020; Lee & Lai, 2020; Lyu et al., 2022; Zhang & Guo, 2025; Arkushin et al., 2025), and customized product design (Bai et al., 2021; Plesh et al., 2023). In the context of eyeglass try-on, most methods adopt predefined 3D eyeglass models and composite them onto faces (Li & Yang, 2011; Yuan et al., 2011; Niswar et al., 2011; Huang et al., 2012; 2013; Tang et al., 2014). Recent techniques simulate advanced physical effects, such as refraction (Zhang et al., 2017), or utilize multi-view data for realistic avatar reconstruction (Li et al., 2023). Though effective, they constrain users to predefined shapes and styles. Compared to these prior works, our work addresses the problem of adding and transferring arbitrary reference eyeglasses to facial videos from a single eyeglass reference image, without relying on predefined 3D models or extensive multi-view datasets.

**Facial video editing** Facial video editing has been explored through latent-space manipulation (Shen et al., 2020; Yao et al., 2021; Patashnik et al., 2021) and temporal consistency techniques such as smoothing and optical flow (Tzaban et al., 2022; Alaluf et al., 2022; Xu et al., 2022), often built on StyleGAN (Karras et al., 2019; 2020b). More recent models, including StyleGANEX (Yang et al., 2023b) and FED-NeRF (Zhang et al., 2024), reduce reliance on cropping and alignment. Kim et al. (2023) introduces a DiffAE-based method that significantly improves reconstruction quality but remains limited to classifier- or text-guided editing. S3Editor (Wang et al., 2024a) offers a model-agnostic self-training framework for semantic disentanglement but similarly lacks fine-grained control. Our work adopts DiffAE (Preechakul et al., 2022) to ensure reconstruction fidelity while enabling reference-guided semantic editing for facial videos.

**Inpainting-based image and video editing** Inpainting-based editing is a common strategy for object insertion, where masked regions are filled with plausible content guided by a reference. Recent diffusion-backboned models (Song et al., 2023; Yang et al., 2023a; Chen et al., 2024b; Song et al., 2024; Chen et al., 2024a) achieve good semantic coherence, but when applied frame by frame, they struggle with temporal consistency. Training-free variants such as TF-ICON (Lu et al., 2023) reduce the need for fine-tuning but still require accurate masks. Video-oriented extensions, including VideoAnyDoor (Tu et al., 2025) and VACE (Jiang et al., 2025), improve temporal alignment but remain mask-dependent. This mask-based formulation is reasonable for generic object insertion but is ill-suited for eyeglass transfer. As inpainting restricts editing to masked regions, these methods treat the eyeglass area independently of the surrounding face, which often weakens harmonization with facial geometry and identity. Our approach differs by integrating reference features directly into the semantic encoder, allowing the model to adapt eyeglasses to pose and

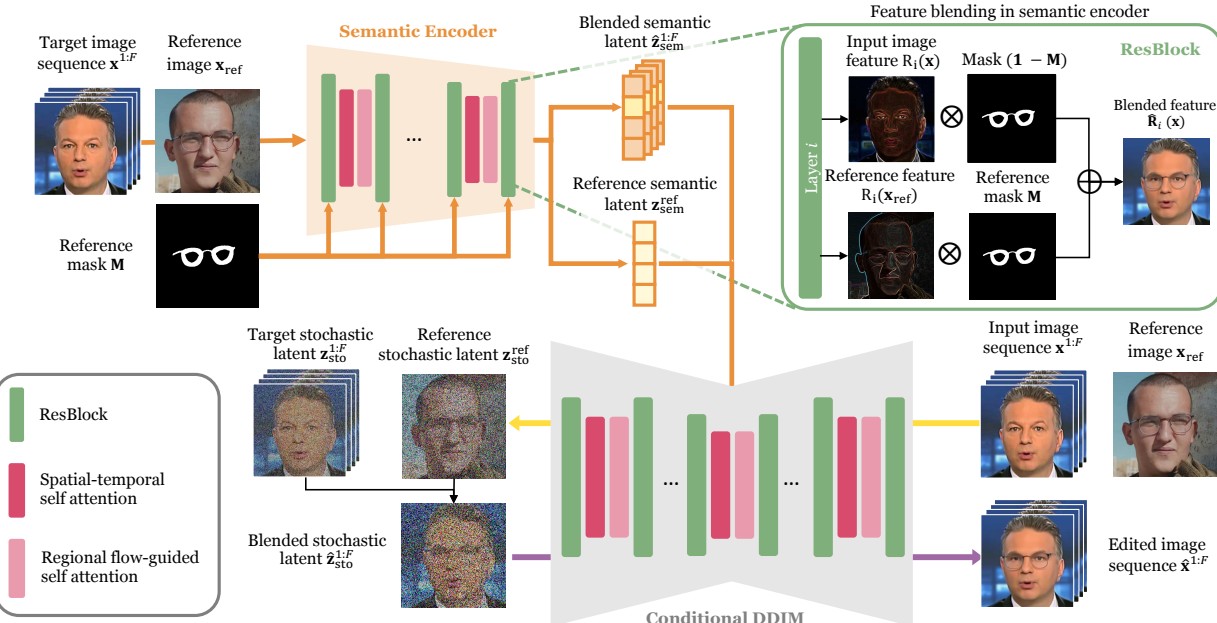

Figure 2: **Overview of our FreeEyeglass pipeline.** Given a target video (*i.e.*, target image sequence) and a reference image of desired eyeglasses, we blend the features of the reference eyeglasses and the target image sequence to obtain a blended semantic latent sequence. We then compute the stochastic latent sequences with the input images and semantic latent sequences through conditional DDIM inversion and construct a blended stochastic latent sequence. Using the blended stochastic latent and semantic latent sequences, we can obtain the final edited image sequence with our desired eyeglasses semantically and naturally placed through condition DDIM sampling.

context without relying on explicit target masks, which leads to more natural and temporally consistent results in video.

**General text-based video editing** Recent text-based video editing methods (Cong et al., 2024; Kara et al., 2024; Li et al., 2024; Yang et al., 2024; Wang et al., 2025) enable high-level video manipulation via language prompts. However, these approaches struggle with fine-grained semantic control, particularly in tasks like eyeglass placement, due to the inherent ambiguity in textual descriptions. Our work adopts a reference-based approach for precise and consistent semantic editing across video frames.

## 3 Method: FreeEyeglass

Figure 2 illustrates the overview of our FreeEyeglass pipeline. In this section, we first recap the Diffusion Autoencoder (DiffAE) (Preechakul et al., 2022) and explain how we semantically place the eyeglasses with a pretrained DiffAE.

### 3.1 Preliminary: Diffusion Autoencoder (DiffAE)

Our method is built on DiffAE proposed in Preechakul et al. (2022). In DiffAE, given an input image $\mathbf{x}$, the semantic encoder $\text{Enc}(\mathbf{x})$ maps it to a semantically meaningful latent $\mathbf{z}_{\text{sem}}$. A stochastic latent $\mathbf{z}_{\text{sto}}$, which captures the remaining stochastic details to achieve a near-perfect reconstruction, is then computed through the deterministic DDIM inversion (Song et al., 2021) using a conditional diffusion model $\text{DDIM}(\mathbf{z}_{\text{sto}}, \mathbf{z}_{\text{sem}})$. Taking $\mathbf{z}_{\text{sto}}$ and $\mathbf{z}_{\text{sem}}$ as input, the conditional diffusion model $\text{DDIM}(\mathbf{z}_{\text{sto}}, \mathbf{z}_{\text{sem}})$ reconstructs an image $\hat{\mathbf{x}}$ through the generative DDIM process. A typical DiffAE model uses a U-Net architecture (Ronneberger et al., 2015) similar to the diffusion model proposed in Dhariwal & Nichol (2021), which consists of multiple ResBlocks (He et al.,

2016) and self-attention blocks (Vaswani et al., 2017). The semantic encoder shares the same architecture as the U-Net encoder and conditions the diffusion U-Net by adaptive group normalization (Dhariwal & Nichol, 2021; Huang & Belongie, 2017).

## 3.2 Feature Blending for Eyeglass Transfer

The semantic encoder $\mathrm{Enc}(\cdot)$ is designed to encode the semantic information of the input image. In our case, the input images are the aligned faces from video frames. The semantic encoder maps these input images into a latent space that captures high-level semantic features. Our approach involves blending the feature maps of the target frames with those of the reference eyeglasses at each ResBlock of the semantic encoder. By doing so, we aim to incorporate the semantics of the reference eyeglasses into the semantic latent of the target frame. The blending of feature maps is performed using a binary mask at each ResBlock. We represent each ResBlock $i$ as a function $\mathbf{R}_i(\mathbf{x})$. Let a target frame be $\mathbf{x}$, a reference image be $\mathbf{x}_{\mathrm{ref}}$, and let $\mathbf{M} \in \{0,1\}^{h_i \times w_i}$ be the binary mask indicating the eyeglass region. Here, $h_i$ and $w_i$ are the spatial dimensions of the feature map $\mathbf{R}_i(\cdot)$, and the mask is bilinearly downsampled to match this resolution. We compute the blended feature map $\hat{\mathbf{R}}_i(\mathbf{x})$ as follows:

$$\hat{\mathbf{R}}_i(\mathbf{x}) = \mathbf{M} \odot \mathbf{R}_i(\mathbf{x}_{\mathrm{ref}}) + (\mathbf{1} - \mathbf{M}) \odot \mathbf{R}_i(\mathbf{x}), \tag{1}$$

where $\odot$ denotes element-wise multiplication. The mask $\mathbf{M}$ ensures that the features from the reference eyeglasses are only blended into the specified regions of the target frame. Importantly, although Eq. (1) applies a spatial mask, the blending is performed in the *semantic latent space* rather than in pixel space. The encoded features are progressively aggregated and compressed into a global latent representation, which is not strictly tied to the spatial geometry of the reference image. During decoding, DiffAE reconstructs the image under a strong facial prior, enforcing consistency with the target facial structure. This results in a non-rigid integration of reference features, enabling implicit geometric adaptation of the transferred eyeglasses to the target face.

After the feature blending process, we obtain a new semantic latent $\hat{\mathbf{z}}_{\mathrm{sem}}$ that merges the semantic information from the target frame $\mathbf{x}$ and the reference image $\mathbf{x}_{\mathrm{ref}}$. While integrating the reference eyeglasses into the semantic latent allows the model to place eyeglasses on the target image semantically, this operation can introduce local noise and boundary artifacts due to feature mismatches between the target and reference features. To mitigate these artifacts, we also blend the stochastic latent codes, which capture residual information from the input image $\mathbf{x}$ that is not represented in the semantic latent $\hat{\mathbf{z}}_{\mathrm{sem}}$. By mixing the stochastic latent of the reference image latent $\mathbf{z}_{\mathrm{sto}}^{\mathrm{ref}}$ with that of the target image $\mathbf{z}_{\mathrm{sto}}$, we smooth the transition between reference and target features and suppress boundary inconsistencies. We construct a stochastic-latent blending mask $\tilde{\mathbf{M}}$ by combining the original reference mask $\mathbf{M}$ and its Gaussian-blurred version $\mathrm{Blur}(\mathbf{M})$:

$$\tilde{\mathbf{M}} = \beta \mathbf{M} + \gamma \, \mathrm{Blur}(\mathbf{M}), \tag{2}$$

where $\beta \geq 0$ and $\gamma \geq 0$ are scalar parameters that control the intensity and smoothness of the blending between the stochastic latent codes. We clamp the $\tilde{\mathbf{M}}$ values to ensure they lie within the $[0,1]$ range. We smooth only the border of the blending mask, keeping the interior binary so that high-frequency details are preserved. Using the constructed mask $\tilde{\mathbf{M}}$, we perform alpha blending of the stochastic latent codes of the target and reference images:

$$\hat{\mathbf{z}}_{\mathrm{sto}} = (1 - \tilde{\mathbf{M}}) \odot \mathbf{z}_{\mathrm{sto}} + \tilde{\mathbf{M}} \odot \mathbf{z}_{\mathrm{sto}}^{\mathrm{ref}}, \tag{3}$$

where $\odot$ denotes element-wise multiplication. Using the blended stochastic latent $\hat{\mathbf{z}}_{\mathrm{sto}}$ as the starting noise of the DDIM sampling, we ensure that the generated image not only semantically includes the eyeglasses but also preserves their specific shape and color, improving the preservation of fine details in the edited images.

## 3.3 Self-attention for Temporal Consistency

Building on our success in achieving realistic and semantically consistent transfer of eyeglasses in images, we extend our method to facial videos. Video editing introduces the critical challenge of maintaining temporal consistency, as humans are highly sensitive to discrepancies across frames. Naively applying our image-based method frame by frame results in the eyeglasses appearing slightly different in each frame, leading to a

jarring, flickering effect. As illustrated in Fig. 2, we extend the DiffAE-based architecture for video editing by incorporating 3D convolutions and two self-attention layers: a spatial-temporal self-attention layer and a regional flow-guided self-attention layer to enhance temporal consistency.

**Extending DiffAE for video editing** The original U-Net architecture in DiffAE consists of a series of 2D convolutional residual blocks and spatial self-attention blocks. To adapt this architecture for video editing, we inflate the 2D U-Net into a pseudo-3D U-Net to accommodate the temporal dimension, similar to previous works (Cong et al., 2024; Wu et al., 2023). Specifically, we replace each 2D convolutional layer with a pseudo-3D convolutional layer. For a $3 \times 3$ kernel, we adjust it to a $1 \times 3 \times 3$ kernel. We also expand the spatial self-attention blocks to spatial-temporal self-attention blocks, often used in video diffusion models, by using features from the entire video as queries $\mathbf{Q}$, keys $\mathbf{K}$, and values $\mathbf{V}$. However, applying full spatial-temporal self-attention can lead to undesired modifications in regions outside the eyeglasses area as the model attends to irrelevant embeddings. Thus, we introduce a regional self-attention mechanism that uses optical flows to mitigate this problem.

**Regional self-attention with optical flows** We aim to insert eyeglasses without altering other facial details. To this end, we employ a regional self-attention strategy. While similar local attention has been explored in prior works (Zhang et al., 2022; Cong et al., 2024), we adapt it to our framework by restricting attention to the eyeglass region. In the target video, we define a rough, predefined region of interest (ROI) of the eyeglasses area on the aligned face, computed as the bounding box of the aligned reference mask. We denote the set of all pixels in the ROI bounding box as $\mathcal{C}_{\mathrm{roi}}$, which are appropriately downsampled to fit the spatial dimension of the self-attention block.

We adapt FLATTEN's flow-guided temporal attention to trajectories that pass through our predefined ROI. Instead of attending to the full frame, it attends only to tokens along these ROI-related trajectories, guided by the estimated optical flow. For a video sequence of length $S$, we can obtain a trajectory $\mathcal{T}$ for a pixel in coordinates $(x_0, y_0)$ in the first frame by deriving its coordinates in all subsequent frames as

$$\mathcal{T} = \{(x_0, y_0), ..., (x_s, y_s), ..., (x_S, y_S)\}. \tag{4}$$

We only consider trajectories for which some or all coordinates satisfy $(x_s, y_s) \in \mathcal{C}_{\mathrm{roi}}$, and denote them as $\{\mathcal{T}\}_{\mathrm{roi}} \subset \{\mathcal{T}\}$.

We compute the self-attention of a selected motion trajectory $\mathcal{T} \in \{\mathcal{T}\}_{\mathrm{roi}}$ in a similar manner to FLAT-TEN (Cong et al., 2024). Let the query $\mathbf{Q}$ of a pixel $(x_s, y_s)$ of frame $s$ be the embeddings $\mathbf{h}(x_s, y_s)$. The corresponding embeddings of the remaining coordinates in the same trajectory, $\mathcal{T}^- = \mathcal{T} - (x_s, y_s)$ are concatenated as

$$\mathbf{H}(\mathcal{T}^-) = [..., \mathbf{h}(x_{s-1}, y_{s-1}), \mathbf{h}(x_{s+1}, y_{s+1}), ...]. \tag{5}$$

Regional optical-flow-guided self-attention is thus computed using the dimensionality of the embeddings $d$ as

$$\mathbf{Q} = \mathbf{h}(x_s, y_s), \quad \mathbf{K} = \mathbf{V} = \mathbf{H}(\mathcal{T}^-), \quad \mathrm{RA}(\mathbf{Q}, \mathbf{K}, \mathbf{V}) = \mathrm{Softmax}\left(\frac{\mathbf{Q}\mathbf{K}^\top}{\sqrt{d}}\right)\mathbf{V}. \tag{6}$$

By constraining the attention to the eyeglasses region and guiding it with optical flow, we ensure that each embedding in our ROI attends only to its temporally corresponding embeddings across frames. This approach improves the temporal consistency of the eyeglasses while preventing unwanted artifacts in unrelated areas.

### 3.4 Overall Video Editing Pipeline

We adopt a video editing pipeline proposed in Yao et al. (2021), which comprises three main stages: (1) face alignment and cropping, (2) latent feature encoding, manipulation, and decoding, and (3) unalignment and merging of edited frames into the original video. Given an input video sequence $\mathbf{v} \in \mathbb{R}^{1 \times C \times F \times H \times W}$, where $C$ is the number of channels, $F$ is the number of frames, and $H$, $W$ denote the height and width, respectively, we first align and square-crop the face regions in each frame and obtain a cropped sequence $\mathbf{x}_{1:F}$ Similarly, we process a reference eyeglasses image to obtain its aligned and cropped face region $\mathbf{x}_{\mathrm{ref}}$. This alignment

ensures that the facial features are spatially consistent across frames and with the reference. To enhance temporal consistency in long video sequences that cannot be input at once, we process the video in batches and sample two neighboring frames, one immediately before and one immediately after each batch. These neighboring frames provide additional temporal context, allowing for smoother transitions between batches. The cropped face sequences $\mathbf{x}_{1:F}$ and the cropped reference image $\mathbf{x}_{\text{ref}}$ are then concatenated as $\mathbf{x}_{1:F,\text{ref}}$ and passed through our semantic encoder $\text{Enc}(\mathbf{x}_{1:F,\text{ref}})$. We apply feature blending inside the semantic encoder and obtain the blended semantic latent codes $\hat{\mathbf{z}}_{\text{sem}}^{1:F}$, and the semantic latent for reference $\mathbf{z}_{\text{sem}}^{\text{ref}} = \text{Enc}(\mathbf{x}_{\text{ref}})$.

Next, we obtain the stochastic latent codes $\mathbf{z}_{\text{sto}}^{1:F}$ and $\mathbf{z}_{\text{sto}}^{\text{ref}}$ via conditional DDIM inversion using $\hat{\mathbf{z}}_{\text{sem}}^{1:F}$ and $\mathbf{z}_{\text{sem}}^{\text{ref}}$. We construct the blended stochastic latent codes $\hat{\mathbf{z}}_{\text{sto}}^{1:F}$ using $\mathbf{z}_{\text{sto}}^{1:F}$ and $\mathbf{z}_{\text{sto}}^{\text{ref}}$ by Equation (3). The edited frames are then generated by passing the blended stochastic latent codes $\hat{\mathbf{z}}_{\text{sto}}^{1:F}$ and semantic latent codes $\hat{\mathbf{z}}_{\text{sem}}^{1:F}$ through our conditional DDIM model. To ensure temporal coherence, we incorporate regional optical-flow-guided self-attention in all self-attention blocks, applying it exclusively to the target video sequence. Finally, we unalign the edited face regions and paste them back into the original video.

# 4 Experiments

We evaluate our method with various baselines to confirm its effectiveness. We describe our implementation details and report further experiments in the appendix.

## 4.1 Settings

**Datasets** Since there is no existing dataset tailored for evaluating semantic eyeglass transfer, we construct a benchmark from CelebV-HQ (Zhu et al., 2022). For target facial videos, we randomly select videos without visible glasses that are at least 120 frames long, *i.e.*, 4 seconds in 30 fps, and use the first 120 consecutive frames of these videos as target facial videos. To obtain reference eyeglasses, we select videos with clearly visible glasses and apply the following quality filters: an average VSFA score (Li et al., 2019) above 0.8,

Table 1: Baseline requirements. Most existing approaches depend on training and/or explicit masks. In contrast, **our method is training-free and target-mask-free**, requiring only a reference eyeglass and the target video.

| Method | Target Mask | Training | Inputs | Backbone |
|---|---|---|---|---|
| TF-ICON (Lu et al., 2023) | Yes | No | Image + Ref + Mask | Diffusion |
| Paint-by-Example (Yang et al., 2023a) | Yes | Yes | Image + Ref + Mask | Diffusion |
| ObjectStitch (Song et al., 2023) | Yes | Yes | Image + Ref + Mask | Diffusion |
| AnyDoor (Chen et al., 2024b) | Yes | Yes | Image + Ref + Mask | Diffusion |
| MimicBrush (Song et al., 2024) | Yes | Yes | Image + Ref + Mask | Diffusion |
| OmniTry (Feng et al., 2025) | No | Yes | Image + Ref | Diffusion |
| DVAE (Kim et al., 2023) | No | Yes | Video + Classifier/Text | DiffAE |
| VideoEditGAN (Tzaban et al., 2022) | No | Yes | Video + Classifier | GAN |
| FLATTEN (Cong et al., 2024) | No | Yes | Video + Text | Diffusion |
| RAVE (Kara et al., 2024) | No | Yes | Video + Text | Diffusion |
| VidToMe (Li et al., 2024) | No | Yes | Video + Text | Diffusion |
| FRESCO (Yang et al., 2024) | No | Yes | Video + Text | Diffusion |
| RF-Solver-Edit (Wang et al., 2025) | No | Yes | Video + Text | Diffusion |
| VACE (Jiang et al., 2025) | Yes | Yes | Video + Ref + Text + Mask | Diffusion |
| **FreeEyeglass (Ours)** | **No** | **No** | Video + Ref | DiffAE |

Ref: Reference Image

mean luminance $> 90$, head pitch and yaw within $\pm 15°$, and unique identity per sample. After filtering, we randomly selected 100 unique pairs of glasses, comprising 75 eyeglasses and 25 sunglasses, to serve as reference eyeglasses. Finally, we prepare eyeglass and sunglass masks using Grounded-SAM (Ren et al., 2024). Specifically for eyeglass masks, we refine them by subtracting the segmented eye region from the segmented eyeglasses region. We use the aligned mask's bounding box as the ROI for our regional flow-guided self-attention. In total, we construct 100 pairs of target facial videos and reference glasses for our main experiment. We also render a small-scale CG face dataset of four target identities, each with one pair of eyeglasses and one pair of sunglasses. We render eight ground-truth videos in total for the target identities. This CG dataset is used only for supplementary pixel-level evaluation, while the main benchmark is built from the CelebV-HQ videos.

**Baselines** We compare against a wide range of state-of-the-art image and video editing methods. Table 1 summarizes the requirements of all baselines in terms of training, masks, and input modalities. For reference-based image editing, we include TF-ICON, Paint-by-Example, ObjectStitch, AnyDoor, and MimicBrush, which insert a reference object into a background image through inpainting. We also evaluate OmniTry, a concurrent method targeting wearable object try-on. As these are image editing methods, we apply them frame by frame to videos.

For video editing, we consider both facial and general text-guided approaches. Facial video editing baselines include Diffusion Video Autoencoder (DVAE) and VideoEditGAN. Text-guided video editing baselines include FLATTEN, RAVE, VidToMe, FRESCO, and RF-Solver-Edit with HunyuanVideo (Kong et al., 2024) as backbone. We further include VACE 1.3B, a concurrent multi-modal video editing framework that requires

Table 2: **Quantitative results** on our evaluation benchmark. The top rows show the methods for image editing, while the bottom rows are for video editing. Eye preservation is evaluated only on eyeglasses. Values closer to 1.0 indicate better in TL-ID and TG-ID. **Bold** and underline indicate the best and second-best results. While our method achieves state-of-the-art results in many cases, it yields a remarkably better *Unified* score $S_{edit}$ that balances identity preservation and editing quality.

| Methods | Editing Fidelity | | | | Temporal Consistency | | | | Eye Preservation | | ID ↑ | $S_{edit}$ ↑ |
|---|---|---|---|---|---|---|---|---|---|---|---|---|
| | $FID_{CLIP}$ ↓ | FVD ↓ | CLIP-I ↑ | DINO-I ↑ | CLIP-F ↑ | $E_{warp}$ ↓ | TL-ID − | TG-ID − | $LPIPS_{eye}$ ↓ | $SSIM_{eye}$ ↑ | | |
| Object Stitch (Song et al., 2023) | 11.626 | 294.12 | 0.882 | 0.624 | 0.957 | 0.0182 | 0.957 | 0.887 | 0.183 | 0.814 | 0.593 | 48.336 |
| TF-ICON (Lu et al., 2023) | 26.394 | 2382.6 | 0.802 | 0.441 | 0.868 | 0.0462 | 0.222 | 0.216 | 0.214 | 0.804 | 0.110 | 17.457 |
| Paint-by-Example (Yang et al., 2023a) | 12.915 | 295.03 | 0.877 | 0.650 | 0.957 | 0.0182 | 0.956 | 0.884 | 0.191 | 0.810 | 0.568 | 48.090 |
| Anydoor (Chen et al., 2024b) | 18.469 | 1007.7 | 0.850 | 0.517 | 0.949 | 0.0279 | 0.779 | 0.748 | 0.242 | 0.781 | 0.559 | 30.452 |
| MimicBrush (Chen et al., 2024a) | 13.399 | 362.31 | **0.909** | **0.739** | 0.959 | 0.0188 | 0.960 | 0.899 | 0.211 | 0.788 | 0.582 | 48.334 |
| OmniTry (Feng et al., 2025) | 11.776 | 415.29 | 0.857 | 0.588 | 0.952 | 0.0173 | 0.892 | 0.684 | 0.183 | 0.813 | 0.511 | 49.670 |
| VideoEditGAN (Xu et al., 2022) | 10.443 | 382.93 | 0.846 | 0.547 | 0.958 | 0.0167 | 0.964 | 0.930 | 0.149 | 0.840 | 0.642 | 50.770 |
| DVAE Classifier (Kim et al., 2023) | 13.624 | 1190.9 | 0.819 | 0.433 | 0.951 | 0.0169 | 0.788 | 0.882 | **0.124** | **0.869** | **0.732** | 48.396 |
| FLATTEN (Cong et al., 2024) | 52.022 | 1522.7 | 0.780 | 0.300 | 0.952 | 0.0154 | 0.860 | 0.718 | 0.175 | 0.839 | 0.381 | 50.809 |
| RAVE (Kara et al., 2024) | 43.152 | 1035.0 | 0.833 | 0.526 | **0.965** | 0.0240 | 0.916 | 0.850 | 0.215 | 0.797 | 0.150 | 34.085 |
| VidToME (Li et al., 2024) | 36.361 | 960.97 | 0.828 | 0.499 | 0.959 | 0.0201 | 0.931 | 0.802 | 0.200 | 0.805 | 0.340 | 41.111 |
| FRESCO (Yang et al., 2024) | 53.046 | 973.35 | 0.787 | 0.486 | 0.952 | 0.0262 | 0.859 | 0.821 | 0.161 | 0.827 | 0.142 | 30.086 |
| RF-Solver-Edit (Wang et al., 2025) | 21.507 | 485.35 | 0.833 | 0.492 | 0.944 | 0.0205 | 0.946 | 0.894 | 0.164 | 0.838 | 0.363 | 40.627 |
| VACE (Jiang et al., 2025) | 9.947 | 223.57 | 0.858 | 0.634 | 0.961 | 0.0167 | **0.974** | **0.942** | 0.163 | 0.836 | 0.665 | 53.136 |
| **FreeEyeglass (Ours)** | **9.839** | **206.37** | 0.865 | 0.542 | 0.962 | **0.0152** | 0.969 | 0.885 | 0.124 | 0.868 | 0.622 | **56.976** |

text prompts, target masks, and reference images. We could not run the 14B variant of VACE due to memory limits on A100 80GB GPUs. For text-based baselines, we use GPT-4o (API version 2024-05-13) (OpenAI, 2024) to generate fine-grained descriptions of reference eyeglasses. Implementation details and prompt preparation are provided in the appendix. We use the implementations and pretrained models provided by the authors for all baselines except FLATTEN to generate edited videos. Due to GPU memory constraints, FLATTEN cannot process a video of 120 frames simultaneously, so we split each video into three chunks.

**Evaluation metrics**   Since our benchmark lacks ground truth, we assess results using four categories of metrics: *editing fidelity*, *temporal consistency*, *eye preservation*, and *identity preservation*.

*Editing fidelity:* We use the Fréchet Inception Distance (FID) computed on CLIP features (Kynkäänniemi et al., 2023) to assess overall perceptual quality and Fréchet Video Distance (FVD) (Skorokhodov et al., 2022) to evaluate video realism. We use all frames in the videos for FVD calculation. We compute CLIP-I and DINO-I scores following Wei et al. (2024), using average cosine similarity between each edited frame and the reference eyeglasses image to assess semantic alignment with the reference. CLIP-I uses CLIP ViT-B/32 (Radford et al., 2021), while DINO-I uses DINOv2 ViT-S/16 (Oquab et al., 2024); in both cases, we align frames and crop the eyeglasses region.

*Temporal consistency:* We compute the average cosine similarity between consecutive frame embeddings using CLIP (CLIP-F) to capture feature-level smoothness, and Warp Error (Lai et al., 2018) that measures the pixel-wise difference between adjacent frames warped using the optical flow from the original video. We also compute TL-ID and TG-ID (Tzaban et al., 2022), which quantify identity preservation across adjacent frames or across all frames, respectively.

*Eye preservation:* We crop the eyeglasses region for all frames and report $LPIPS_{eye}$ and $SSIM_{eye}$ for the eyeglasses regions of source frames and edited frames, similar to Feng et al. (2025). We conduct the evaluation exclusively on 75 eyeglass pairs, as sunglasses typically obscure the eye region.

*Identity preservation:* We evaluate identity preservation using ArcFace (Deng et al., 2019) (ID). For each edited video, we compute the cosine similarity between the normalized ArcFace embeddings of the corresponding frames from the edited and original target videos. We report the mean similarity across all videos.

We also report a *unified evaluation* score $S_{edit}$, proposed by FLATTEN, defined as the ratio CLIP-I/$E_{warp}$, providing a single metric that balances semantic fidelity and temporal consistency. For the CG face dataset, we compute PSNR, MS-SSIM, LPIPS (Zhang et al., 2018), and MSE between ground-truth frames and edited frames generated by ours and the baselines.

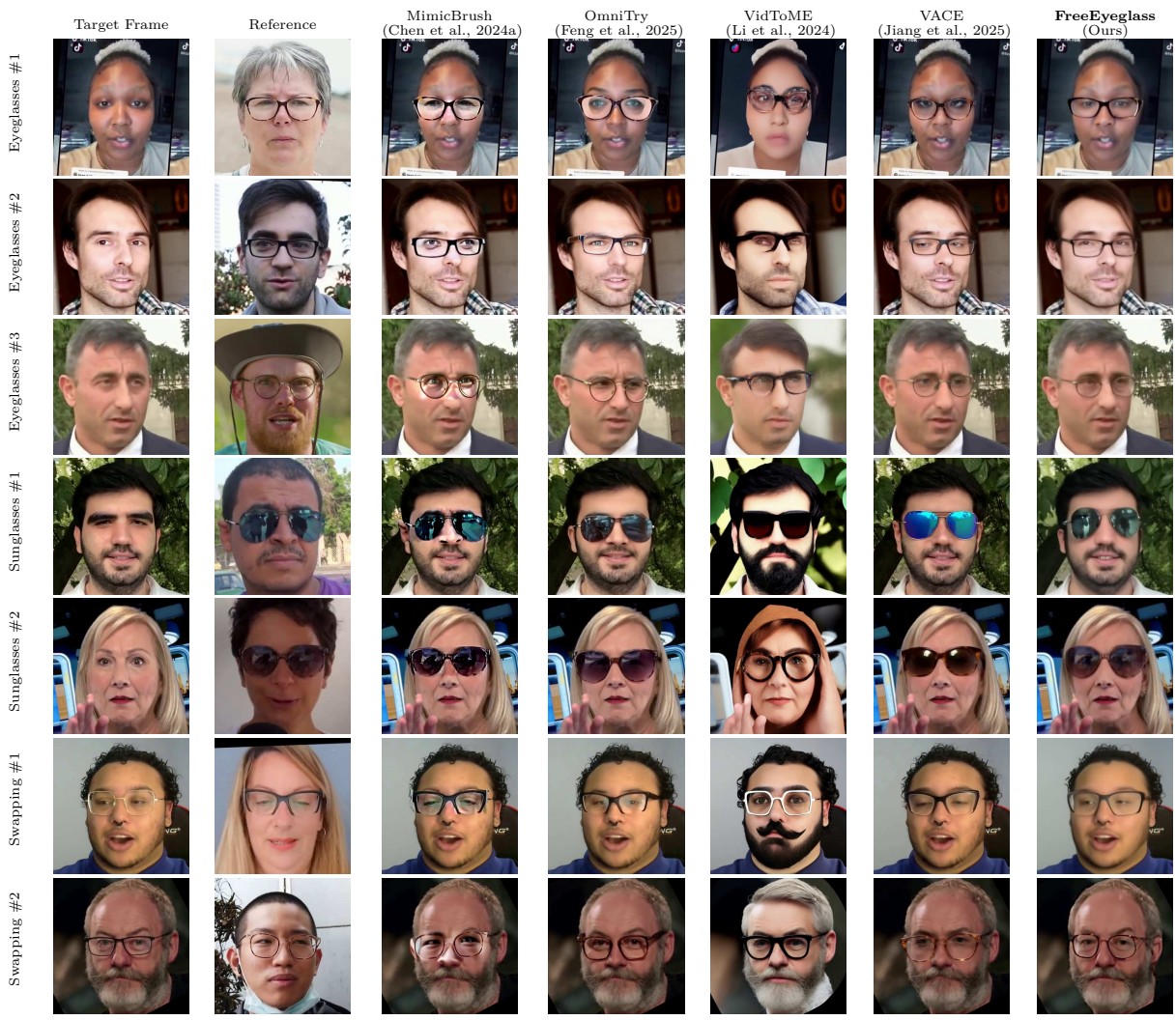

Figure 3: **Visual results** on transferring different eyeglasses compared with our baselines. Our method successfully inserts and swaps eyeglasses from the reference eyeglasses image into the target video, compared to existing baselines. Please refer to the supplementary video for more results.

## 4.2 Main Results

We present our quantitative results in Table 2 and visual comparisons with baselines in Fig. 3. Our method achieves the best unified score $S_{edit}$ among all baselines, reflecting a clear advantage in balancing editing fidelity and temporal consistency. We also obtain strong $FID_{CLIP}$, FVD, and eye preservation scores $LPIPS_{eye}$ and $SSIM_{eye}$, demonstrating our ability to preserve original video content and maintain overall realism. We also report ArcFace identity similarity (ID) to measure identity preservation of edited frames compared to the original target video. Methods such as DVAE-Classifier and VideoEditGAN obtain higher ArcFace scores; however, these approaches largely preserve the original facial appearance and do not reliably perform reference-guided eyeglass transfer. Their higher identity similarity, therefore, partly reflects weaker appearance modification rather than stronger editing performance. Among methods that consistently insert reference eyeglasses, our method achieves identity similarity comparable to strong baselines such as VACE while maintaining stable eyeglass appearance across frames. Importantly, qualitative results show that inpainting-based methods, including the competing approaches, modify or overwrite the eye region, affecting both eye appearance and motion (*e.g.*, blink behavior and gaze direction) and fail to maintain facial identity (See *e.g.* Eyeglass #1 in Fig. 3) In contrast, our method preserves the original eye structure and motion while

transferring the reference eyeglasses. Video editing methods guided by text or classifiers are temporally stable but cannot capture the reference eyeglass style.

We note that the outputs may appear slightly smoother than the original frames. The primary cause is the approximation of stochastic details after semantic editing. Since the blended semantic representation does not correspond to an observed real image, its exact stochastic latent is unavailable and is approximated by blending the inverted stochastic latents of the target and reference. This approximation may not perfectly match the edited semantic representation, leading to a loss of high-frequency detail. Secondary factors include the $(256 \times 256)$ reconstruction resolution of the DiffAE backbone and the boundary smoothing used during stochastic-latent blending to suppress seams and compositing artifacts. In the current framework, this effect can be reduced by weakening the smoothing or using a narrower blending boundary, although this introduces a trade-off with boundary artifacts. More generally, higher-resolution DiffAE backbones and post-editing or latent-space detail refinement could further improve sharpness. We provide a controlled analysis of these factors in the supplementary material.

We adopt DVAE with classifier guidance as our main DVAE baseline in Table 2 following its original video editing setup. We further analyze DVAE with text guidance and the original image-based DiffAE with classifier guidance in Sec. 4.6 to understand how the DiffAE backbone behaves under different guidance types. We also include a supplementary video for full comparison.

Our method scores slightly lower on CLIP-I and DINO-I, as these metrics favor methods that closely match the reference, yielding high scores when directly copying eyeglasses. Inpainting-based baselines, including Object Stitch, Paint-by-example, Anydoor, and MimicBrush, use CLIP or DINO features during training, which provides them an advantage. In contrast, our method has no prior exposure to these embeddings. Finally, our framework requires neither per-frame target-mask annotations nor model training. Despite being training-free and target-mask-free, it delivers results that are competitive with, or superior to, those of training-based state-of-the-art methods.

## 4.3 Evaluation on CG-rendered Ground Truth

We present our quantitative results on CG-rendered scenes in Table 3. Our method outperforms all baselines on PSNR, SSIM, and MSE, and achieves competitive performance on LPIPS, demonstrating superior fidelity across both pixel-level and perceptual metrics. Text-based video editing often over-modifies the appearance and style of the video when applied to local regions, *i.e.*, eyeglass transfer, thus leading to unsatisfactory results. This emphasizes the importance of reference-based editing for achieving precise semantic control. We provide a detailed breakdown of per-video results in the supplementary materials.

## 4.4 Robustness and Generalization Analysis

**Robustness to lighting variations**  Our method remains stable under strong illumination mismatches between the reference eyeglasses and the target video frames. As in Fig. 4, it preserves correct eyeglass geometry and placement when transferring from a front-lit indoor reference to targets captured under bright outdoor sunlight, strong backlighting, and dim indoor side-lit. Across all cases, the transferred glasses integrate cleanly into each scene without artifacts.

**Robustness to pose change**  We analyze FreeEyeglass's performance across varying head yaw angles. As

Table 3: **Quantitative results** on the CG face dataset. Our method outperforms on most metrics. **Bold** and underline indicate the best and second-best results.

| Methods | PSNR ↑ | SSIM ↑ | LPIPS ↓ | MSE ↓ |
|---|---|---|---|---|
| Object Stitch | 26.495 | 0.956 | 0.0401 | 481.30 |
| TF-ICON | 10.596 | 0.200 | 0.503 | 17617 |
| Paint-by-Example | 26.482 | 0.957 | 0.0392 | 460.58 |
| Anydoor | 24.405 | 0.947 | 0.0431 | 709.85 |
| MimicBrush | 26.527 | 0.964 | **0.0301** | 467.81 |
| OmniTry | 25.251 | 0.945 | 0.0395 | 629.83 |
| DVAE Classifier | 25.626 | 0.942 | 0.118 | 572.16 |
| VideoEditGAN | 24.008 | 0.912 | 0.0715 | 843.26 |
| FLATTEN | 16.857 | 0.780 | 0.212 | 4245.6 |
| RAVE | 15.942 | 0.799 | 0.175 | 6440.2 |
| VidToME | 17.802 | 0.813 | 0.193 | 3411.6 |
| FRESCO | 18.537 | 0.798 | 0.148 | 2779.3 |
| RF-Solver-Edit | 18.548 | 0.595 | 0.431 | 2936.0 |
| VACE | 26.688 | 0.958 | 0.0375 | 463.53 |
| **FreeEyeglass (Ours)** | **28.425** | **0.967** | 0.0387 | **307.21** |

SSIM: MS-SSIM

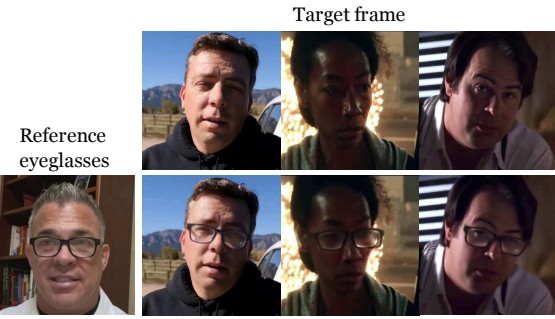

Figure 4: Transferring eyeglasses to target frames under **different illuminations** (*e.g.*, outdoor sunlight, backlit, and side-lit).

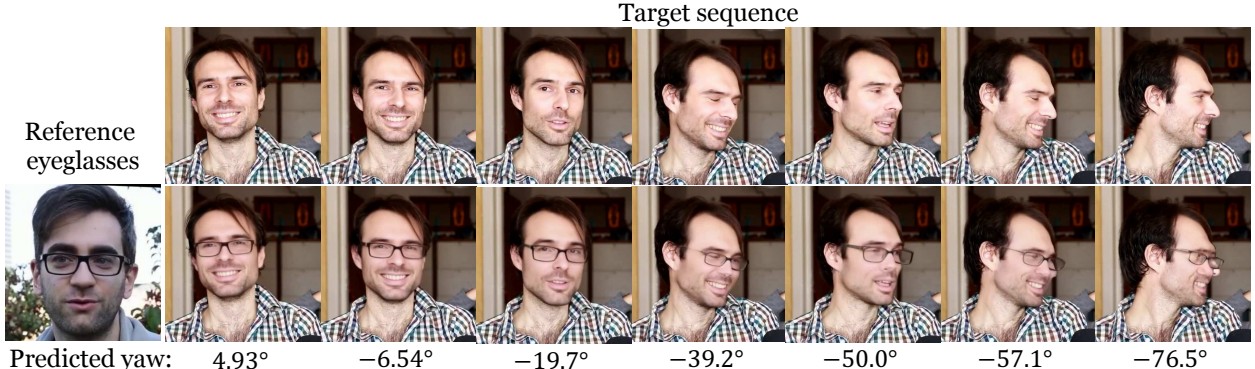

Figure 5: **Eyeglass transfer under head pose variation.** Edited results across increasing yaw angles. Our method remains stable up to roughly ±40–50°, beyond which misalignment and artifacts occur.

shown in Fig. 5, our method remains visually stable under moderate pose variation and continues to place the transferred eyeglasses with plausible geometry up to approximately ±40–50°. Beyond this range, the transfer becomes less reliable due to increased geometric mismatch between the reference and target. We perform a quantitative analysis on editing fidelity across head yaw angles using CLIP-I computed on the eyeglass region. FreeEyeglass shows stable performance across moderate pose variations, with CLIP-I decreasing only slightly from 0.870 at near-frontal views to 0.851 at extreme poses ($> 50°$). MimicBrush and VACE exhibit similar degradation trends under large viewpoint differences. These results suggest that the performance drop at extreme poses reflects the intrinsic difficulty of reference-guided eyeglass transfer rather than a limitation of our feature blending formulation.

**Occlusion handling**   We evaluate FreeEyeglass under several types of partial occlusions. As shown in Fig. 6, the method produces reasonable results in cases involving hair bangs, dynamic hand motion, and rigid objects such as headphones or microphones. Our method preserves the target identity and eyeglass appearance, while maintaining plausible geometry and placement. We attribute this robustness to the strong reconstruction priors of DiffAE, which enable the model to infer a consistent facial structure despite partial visibility.

We further evaluate performance under varying levels of facial occlusion in Table 5. For each frame, the occlusion ratio is estimated as the fraction of non-face pixels inside the convex-hull face region. Frames are then grouped into occlusion bins and evaluated using CLIP-I on the eyeglass region. FreeEyeglass maintains stable editing fidelity under typical occlusion levels ($< 10\%$), with CLIP-I decreasing gradually from 0.869 to 0.838 as occlusion increases. MimicBrush and VACE show comparable trends across occlusion bins. This indicates that moderate occlusions have a limited impact on reference-guided eyeglass transfer, while larger occlusions naturally increase the difficulty of reconstructing the occluded region.

**Generalization to other facial attributes**   Although FreeEyeglass is designed for eyeglass transfer, we push the limits of the framework by applying the same mask-only blending strategy to other local facial attributes, including eyebrows, noses, and mustaches (Fig. 7). Without any model modification, the method can inject these attributes into the target sequence with reasonable geometric alignment. These examples suggest potential applicability to other face-centric attributes. We also explore sequential multi-attribute editing and present the results in Sec. C.6.

## 4.5   Ablation Study

We conduct an ablation study across various settings to demonstrate the effectiveness of our method. We ablate our method by switching off different proposed strategies: (1) w/o feature blending, (2) w/o blended

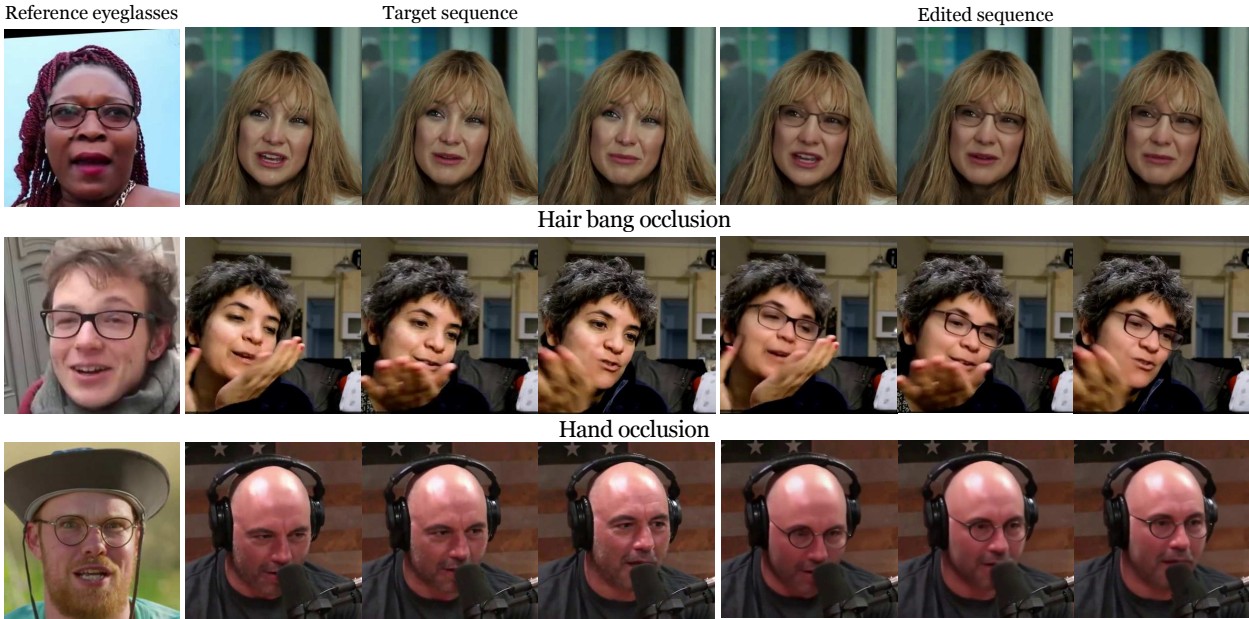

Reference eyeglasses    Target sequence         Edited sequence

Hair bang occlusion

Hand occlusion

Headphone and microphone occlusion

Figure 6: **Occlusion handling results.** FreeEyeglass handles cases when the target face is partially occluded. We present three representative occlusions, including hair bangs, hand motion, and object occlusion over the face, where the transferred eyeglasses remain well-aligned and visually coherent.

| Yaw bin | Frames | CLIP-I↑ | | |
|---|---|---|---|---|
| | | Ours | MimicBrush | VACE |
| $< 10°$ | 4397 | 0.870 | 0.917 | 0.869 |
| $10–20°$ | 3300 | 0.868 | 0.906 | 0.852 |
| $20–30°$ | 1976 | 0.864 | 0.909 | 0.853 |
| $30–40°$ | 980 | 0.854 | 0.898 | 0.864 |
| $40–50°$ | 975 | 0.850 | 0.899 | 0.845 |
| $> 50°$ | 272 | 0.851 | 0.885 | 0.830 |

Table 4: **Pose robustness across head yaw angles.** CLIP-I is computed on the eyeglass crop. FreeEyeglass remains stable across moderate poses, while all methods exhibit similar degradation at extreme viewpoints.

| Occlusion bin | Frames | CLIP-I↑ | | |
|---|---|---|---|---|
| | | Ours | MimicBrush | VACE |
| $0–5\%$ | 4136 | 0.869 | 0.912 | 0.859 |
| $5–10\%$ | 7046 | 0.864 | 0.907 | 0.860 |
| $10–15\%$ | 551 | 0.854 | 0.904 | 0.838 |
| $\geq 15\%$ | 167 | 0.838 | 0.895 | 0.860 |

Table 5: **Occlusion robustness.** CLIP-I is evaluated across facial occlusion levels. FreeEyeglass maintains stable performance under moderate occlusion, with gradual degradation at higher occlusion levels.

stochastic latent $\hat{\mathbf{z}}_{\mathrm{sto}}$, and different self-attention variants, including per-frame, spatial–temporal, flow-guided, and their regional counterparts. Figure 8 shows that feature blending is essential for inserting the reference eyeglasses; the "w/o feat. blend" variant yields a lower FVD simply because it produces almost no edits and therefore remains closer to the target distribution. Using feature blending without stochastic blending yields the most faithful results for the eyeglasses (Table 6), but introduces noticeable noise in Fig. 8. Adding a fixed ROI boundary to per-frame SA or STSA reduces FVD by limiting global interference, but it also lowers CLIP-I because the static ROI often misaligns with the eyeglasses, causing partial corruption of their shapes. In contrast, combining the same ROI boundary with flow-guided attention maintains both semantic correctness and temporal alignment, as it tracks motion trajectories passing through the ROI and keeps the attended features aligned with the moving eyeglasses across frames. We also compare different locations to apply feature blending within DiffAE (see Sec. C.7). Results confirm that applying blending in the semantic encoder yields the best identity-preserving edits, whereas applying it elsewhere causes artifacts or copy-paste effects.

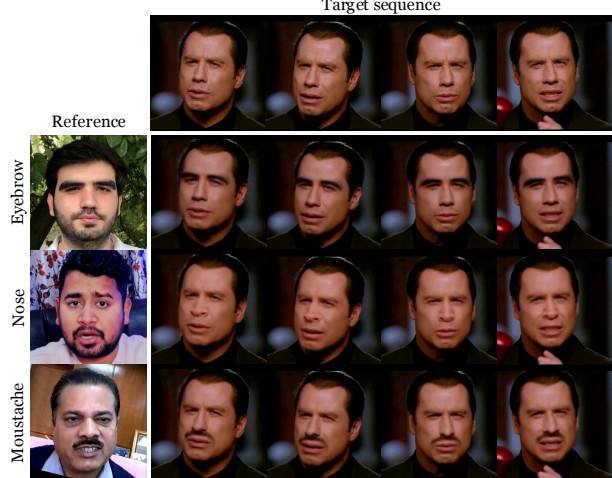

Figure 7: **Transfer of other facial attributes.** FreeEyeglass can transfer other face-centric attributes.

Table 6: **Ablation study** on different components of our method. **Bold** and underline indicate the best and second-best results.

| Settings | FVD↓ | CLIP-I↑ | $E_{warp}$ ↓ | TL-ID− |
|---|---|---|---|---|
| Our full model | 206.37 | 0.865 | 0.0152 | 0.969 |
| w/o feat. blend. | **167.65** | 0.819 | **0.0149** | **0.970** |
| w/o blended $\hat{z}_{sto}$ | 280.73 | **0.868** | 0.0162 | 0.946 |
| Per-frame SA | 226.37 | 0.863 | 0.0151 | 0.968 |
| STSA | 226.51 | 0.864 | 0.0151 | 0.968 |
| FlowSA | 223.61 | 0.864 | 0.0151 | 0.969 |
| Regional Per-frame SA | 215.03 | 0.849 | 0.0152 | 0.966 |
| Regional STSA | 193.68 | 0.851 | 0.0152 | 0.958 |

Feat. blend.: feature blending strategy
SA: self-attention
STSA: spatial-temporal self-attention
FlowSA: flow-guided self-attention

Table 7: **Performance of DiffAE-based methods** on the eyeglass transfer task. DVAE text fails to apply edits despite strong reconstruction-oriented scores.

| Methods | Editing Fidelity | | | | Temporal Consistency | | | | Eye Preservation | | ID ↑ | $S_{edit}$ ↑ |
|---|---|---|---|---|---|---|---|---|---|---|---|---|
| | $FID_{CLIP}$ ↓ | FVD ↓ | CLIP-I ↑ | DINO-I ↑ | CLIP-F ↑ | $E_{warp}$ ↓ | TL-ID − | TG-ID − | $LPIPS_{eye}$ ↓ | $SSIM_{eye}$ ↑ | | |
| DiffAE Classifier (Preechakul et al., 2022) | 8.397 | 295.59 | 0.854 | 0.532 | 0.960 | 0.0167 | 0.962 | 0.866 | 0.166 | 0.850 | 0.687 | 51.072 |
| DVAE Text (Kim et al., 2023) | **6.009** | **152.12** | 0.805 | 0.417 | 0.955 | 0.0160 | 0.960 | **0.916** | **0.0809** | **0.900** | **0.815** | 51.629 |
| DVAE Classifier (Kim et al., 2023) | 13.624 | 1190.9 | 0.819 | 0.433 | 0.951 | 0.0169 | 0.788 | 0.882 | 0.124 | 0.869 | 0.732 | 48.396 |
| **FreeEyeglass (Ours)** | 9.839 | 206.37 | **0.865** | **0.542** | **0.962** | **0.0152** | **0.969** | 0.885 | 0.124 | 0.868 | 0.622 | **56.976** |

Table 8: **Comparison** with the inflated version of MimicBrush (Chen et al., 2024a). Values closer to 1.0 indicate better in TL-ID and TG-ID.

| Methods | Editing Fidelity | | | | Temporal Consistency | | | | Eye Preservation | | ID ↑ | $S_{edit}$ ↑ |
|---|---|---|---|---|---|---|---|---|---|---|---|---|
| | $FID_{CLIP}$ ↓ | FVD ↓ | CLIP-I ↑ | DINO-I ↑ | CLIP-F ↑ | $E_{warp}$ ↓ | TL-ID − | TG-ID − | $LPIPS_{eye}$ ↓ | $SSIM_{eye}$ ↑ | | |
| MimicBrush (Chen et al., 2024a) | 13.399 | 362.31 | 0.909 | 0.739 | 0.959 | 0.0188 | 0.960 | **0.899** | 0.211 | 0.788 | 0.582 | 48.334 |
| Inflated MimicBrush (Chen et al., 2024a) | 13.193 | 648.59 | **0.910** | **0.744** | 0.955 | 0.0226 | 0.867 | 0.843 | 0.211 | 0.788 | 0.577 | 40.229 |
| **FreeEyeglass (Ours)** | **9.839** | **206.37** | 0.865 | 0.542 | **0.962** | **0.0152** | **0.969** | 0.887 | **0.124** | **0.868** | **0.622** | **56.976** |

## 4.6 Analysis of DiffAE-based Editing Methods

We evaluate the DiffAE-based editing mechanisms, original DiffAE with classifier guidance, and DVAE with text guidance in this section. DiffAE with classifier guidance can insert eyeglasses and produce plausible results. However, because the classifier provides only a coarse binary signal (*e.g.*, "wearing glasses"), the generated eyeglasses do not match the reference, as reflected in lower CLIP-I and DINO-I scores. We also observe a weaker temporal consistency. These behaviors reflect the limits of DiffAE's original editing mechanism, as it is not designed for reference-guided or temporally aligned control. DVAE with text guidance almost reconstructs the original frames, yielding deceptively strong scores on metrics that compare against the input (*e.g.*, $FID_{CLIP}$ and FVD), as summarized in Table 7. However, eyeglasses are barely added under the default guidance level (0.5), and stronger guidance forces the faces to collapse (see Fig. 9). In contrast, DVAE with classifier guidance can insert eyeglasses, but the results are less faithful due to the coarse nature of classifier signals. These findings suggest that while DiffAE models possess the capacity for editing, the success of the edit depends strongly on the type and strength of the guidance. Effective and reliable reference-guided control is therefore non-trivial, even with a DiffAE backbone, and motivates our design of FreeEyeglass.

## 4.7 Comparison with Inflated Reference-based Baseline

We further investigate inflating MimicBrush's U-Net (Chen et al., 2024a), the most reference-faithful image editing baseline, from 2D to 3D, following FLATTEN. Our goal is to determine whether simply extending

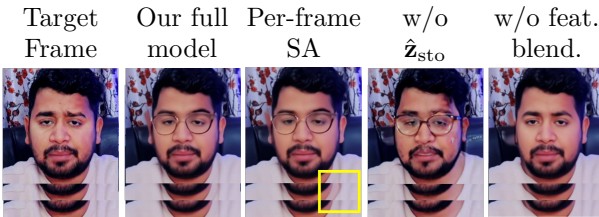

Figure 8: **Visual results** of ablation study. The yellow box denotes the region of temporal inconsistency. Zoom in for a clear comparison.

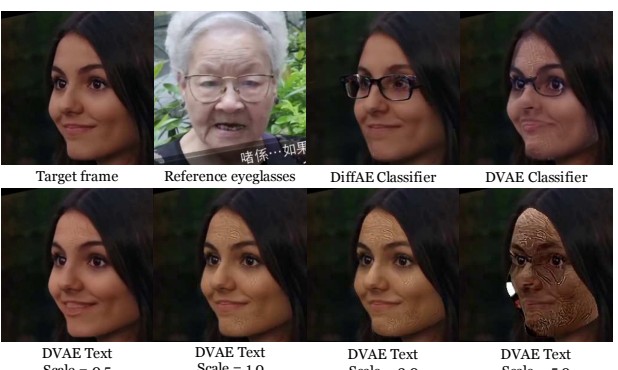

Figure 9: **Visual results** of DiffAE classifier guidance, DVAE classifier, and text guidance.

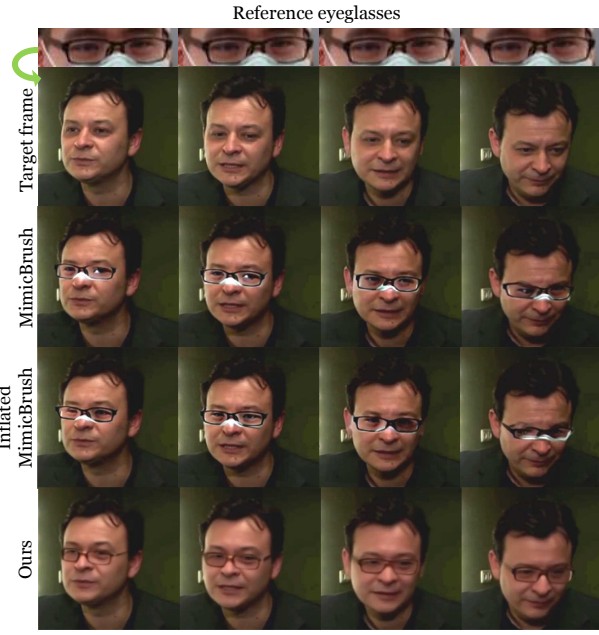

Figure 10: **Visual results** on temporal consistency with MimicBrush and inflated MimicBrush (Chen et al., 2024a).

reference-based image editing methods into the temporal domain straightforwardly improves their performance in video tasks. However, this naive extension worsened temporal consistency and video quality as shown in Table 8 and Fig. 10. These results indicate that temporal adaptation of existing image-based methods is non-trivial and does not necessarily yield better outcomes, likely due to temporal attention mismatches.

## 5 Conclusion

This work targets realistic video eyeglass transfer, where preserving facial identity is critical. To move beyond inpainting-style generation, we propose a training-free, reference-guided framework on reconstruction-oriented DiffAE. Our method blends semantic features into the encoder and uses regional flow-guided attention to propagate eyeglass appearance while maintaining temporal coherence. Experiments show that our method successfully transfers the reference eyeglasses into the target video, creating harmonized results that accurately reflect the original eyeglasses. Our method nevertheless has several limitations. First, it can struggle when swapping eyeglasses that differ substantially in style from those already present in the target or when removing eyeglasses completely (see Sec. G). Fine eyeglass details that are not captured by the semantic encoder remain embedded in the stochastic latent space and are therefore difficult to manipulate. Second, the approximation of stochastic details after semantic editing, together with the limited reconstruction resolution of the DiffAE backbone, can make some outputs appear slightly smoother than the original frames. These issues could be mitigated by improved semantic–stochastic disentanglement, higher-resolution reconstruction backbones, or localized detail refinement. Finally, similar to existing baselines, our method does not explicitly model scene illumination, and lens reflections are inherited from the reference image rather than physically re-rendered. We therefore view reflection-aware generation based on explicit lighting estimation or environment modeling as an important direction for future work.

**Broader impact** This work proposes a method for reference-guided eyeglass transfer in facial videos and has potential societal implications related to privacy, misuse, and fairness. Our experiments use publicly available datasets licensed for use and a private CG dataset created with informed consent, solely for research

purposes. No personally identifying information is collected in our user study, and the authors record no new human data. As the method operates on facial videos, it could be misused to alter a person's appearance without consent or to create deceptive content. Although the method does not aim to modify facial identity or expressions, realistic accessory edits may still mislead viewers if used maliciously. Any real-world deployment should therefore involve appropriate consent, transparency, and safeguards, such as labeling edited content with watermarks or metadata. The method relies on a DiffAE pretrained on the FFHQ dataset. While we introduce no new training data, the learned representation may reflect existing dataset biases, potentially affecting performance across demographic groups. We acknowledge this limitation and view fairness evaluation as an important direction for future work.

## Acknowledgments

This work was partially supported by JSPS KAKENHI Grant JP23H05491 and JST FOREST Grant JPMJFR206F.

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

# Appendix

This appendix provides additional discussion and experimental results. We first present a discussion of related work on text-based editing in Sec. A. We describe further implementation details in Sec. B. We then report extended experiments, including a temporal consistency analysis (Sec. C.1), quantitative analysis for head pose and facial occlusion (Sec. C.2 and Sec. C.3), robustness to mask inaccuracies, viewpoint differences, and sequential attribute editing in Sec. C.4, Sec. C.5, and Sec. C.6. We also include an ablation on the location of feature blending (Sec. C.7), an analysis on perceived soft results (Sec. C.8), per-video results on the CG dataset (Sec. C.9), and an analysis on the evaluation behavior of CLIP-I and DINO-I scores (Sec. C.10). A user study appears in Sec. D, followed by computational analysis in Sec. E and a discussion of limitations in Sec. G. Section F describes how we prepare text prompts for the text-based video-editing baselines. More visual results and comparisons with our baselines are available in the supplementary video. We *strongly* encourage readers to refer to it as well.

## A  Related Work on General Text-based Video Editing

Recent text-based video editing methods (Cong et al., 2024; Kara et al., 2024; Li et al., 2024; Yang et al., 2024; Wang et al., 2025) enable high-level video manipulation via language prompts. However, these approaches struggle with fine-grained semantic control, particularly in tasks like eyeglass placement, due to the inherent ambiguity in textual descriptions. Our work adopts a reference-based approach for precise and consistent semantic editing across video frames.

## B  Implementation Details

We implement our method on the DiffAE FFHQ256 model (Preechakul et al., 2022). As preprocessing, we align all facial videos and reference images similar to the FFHQ dataset (Karras et al., 2019), except we pad black pixels to the missing background and crop them into the resolution of $256^2$. We use RAFT (Teed & Deng, 2020) to estimate the optical flow of each video. For our main experiment, we generate videos of 120 frames at 30 fps using a batch size of 120. In generating a blended mask for stochastic latent, we use a Gaussian blur with kernel size $(25, 25)$ and sigma value $(25, 25)$. We set $\beta = 1, \gamma = 1$ for eyeglasses and $\beta = 0, \gamma = 2$ for sunglasses.

Our system includes several tunable parameters, such as the temporal attention configuration and the number of denoising steps, that can be adjusted to further suppress minor artifacts. In practice, users can choose these settings to match their needs, which trades computational cost for additional visual refinement.

## C  Further Experiments

### C.1  Temporal Consistency Analysis

We compare the temporal consistency of edited videos with OmniTry and VideoEditGAN in Fig. A. The edited eyeglasses of OmniTry change between frames because they do not aim to maintain temporal consistency. While VideoEditGAN generates temporally consistent eyeglasses, it does not match the reference closely due to the language ambiguity involved in text-based control. Additionally, facial expressions are also altered. In contrast, our approach successfully replicates the reference eyeglasses and maintains temporal consistency across frames. This is also reflected

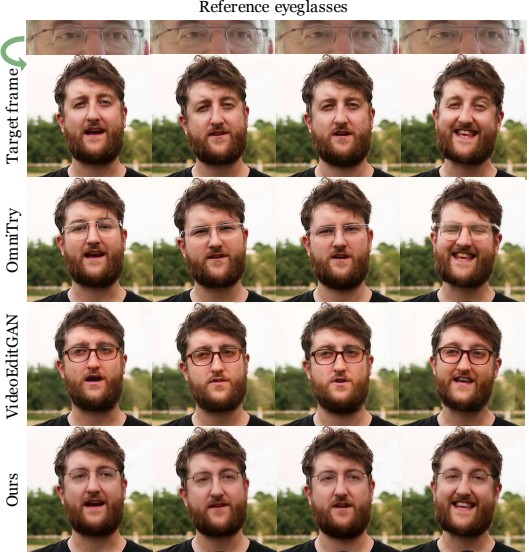

Figure A: **Visual results** on temporal consistency with OmniTry (Feng et al., 2025) and VideoEditGAN (Xu et al., 2022).

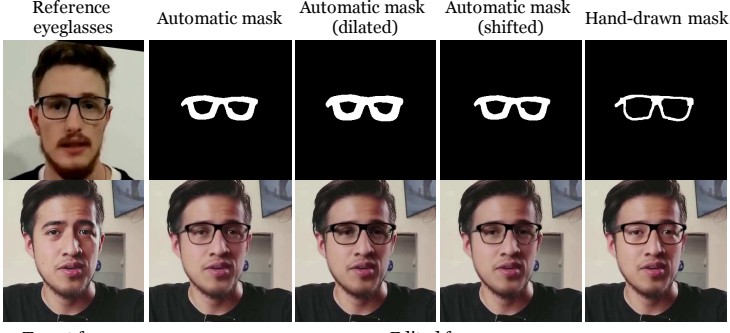

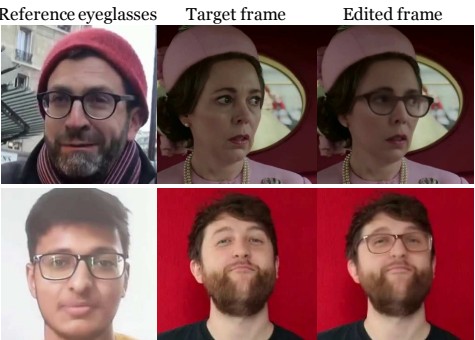

Figure B: We evaluate **the sensitivity of our method to imperfect reference masks**. From left to right: the reference eyeglasses, the automatic mask, a dilated version of the mask (expanded by 2 px), a shifted version (translated 5 px to the right), and a rough hand-drawn mask. The automatic mask is generated by our mask preparation pipeline using Grounded-SAM on the aligned reference frame.

Figure C: **Reference-target viewpoint mismatch.** Examples where the reference eyeglasses are captured from a noticeably different viewpoint than the target frames (top: left-facing reference vs. right-facing target; bottom: pitch-angle mismatch).

| Yaw bin | Frames | Ours | | MimicBrush | | VACE | |
|---|---|---|---|---|---|---|---|
| | | CLIP-I↑ | DINO-I↑ | CLIP-I↑ | DINO-I↑ | CLIP-I↑ | DINO-I↑ |
| $< 10°$ | 4397 | $0.870 \pm 0.046$ | $0.575 \pm 0.142$ | $0.917 \pm 0.034$ | $0.770 \pm 0.109$ | $0.869 \pm 0.042$ | $0.670 \pm 0.140$ |
| $10–20°$ | 3300 | $0.868 \pm 0.035$ | $0.541 \pm 0.125$ | $0.906 \pm 0.031$ | $0.752 \pm 0.110$ | $0.852 \pm 0.045$ | $0.640 \pm 0.130$ |
| $20–30°$ | 1976 | $0.864 \pm 0.032$ | $0.526 \pm 0.126$ | $0.909 \pm 0.023$ | $0.712 \pm 0.095$ | $0.853 \pm 0.046$ | $0.598 \pm 0.106$ |
| $30–40°$ | 980 | $0.854 \pm 0.029$ | $0.485 \pm 0.144$ | $0.898 \pm 0.018$ | $0.690 \pm 0.071$ | $0.864 \pm 0.020$ | $0.605 \pm 0.060$ |
| $40–50°$ | 975 | $0.850 \pm 0.037$ | $0.497 \pm 0.081$ | $0.899 \pm 0.018$ | $0.676 \pm 0.071$ | $0.845 \pm 0.020$ | $0.568 \pm 0.060$ |
| $> 50°$ | 272 | $0.851 \pm 0.043$ | $0.494 \pm 0.098$ | $0.885 \pm 0.018$ | $0.674 \pm 0.071$ | $0.830 \pm 0.020$ | $0.574 \pm 0.060$ |

Table A: Pose-stratified evaluation using CLIP-I and DINO-I computed on the eyeglass crop. All methods show reduced similarity scores at extreme yaw angles, reflecting the increased geometric discrepancy between the reference eyeglasses and the target face.

in our quantitative results, where we achieve the state-of-the-art scores in temporal consistency. Please refer to the supplementary video for a full comparison.

## C.2 Quantitative Analysis under Head Pose Variation

To examine the behavior of reference-guided eyeglass transfer under viewpoint changes, we perform a pose-stratified evaluation. For each frame, the head yaw angle is estimated from facial landmarks, and the frames are grouped into yaw bins. Editing fidelity is evaluated using two reference similarity metrics computed on the eyeglass crop: CLIP-I, which measures semantic similarity to the reference image, and DINO-I, which measures feature similarity using self-supervised visual representations.

Table A summarizes the results for FreeEyeglass and two strong baselines, MimicBrush and VACE. Across moderate yaw ranges, the CLIP-I scores of FreeEyeglass remain highly stable, while DINO-I exhibits a gradual decrease as the viewpoint becomes more oblique. A comparable pattern can also be observed for the baseline methods. These observations indicate that large pose differences increase the difficulty of aligning the transferred eyeglasses with the target face, rather than revealing a failure mode specific to the feature blending mechanism used in FreeEyeglass.

| Occlusion bin | Frames | Ours | | MimicBrush | | VACE | |
|---|---|---|---|---|---|---|---|
| | | CLIP-I↑ | DINO-I↑ | CLIP-I↑ | DINO-I↑ | CLIP-I↑ | DINO-I↑ |
| 0–5% | 4136 | $0.869 \pm 0.041$ | $0.546 \pm 0.114$ | $0.912 \pm 0.034$ | $0.743 \pm 0.109$ | $0.859 \pm 0.042$ | $0.621 \pm 0.140$ |
| 5–10% | 7046 | $0.864 \pm 0.038$ | $0.546 \pm 0.144$ | $0.907 \pm 0.031$ | $0.738 \pm 0.110$ | $0.860 \pm 0.045$ | $0.644 \pm 0.130$ |
| 10–15% | 551 | $0.854 \pm 0.044$ | $0.455 \pm 0.123$ | $0.904 \pm 0.023$ | $0.729 \pm 0.095$ | $0.838 \pm 0.046$ | $0.603 \pm 0.106$ |
| $\geq 15\%$ | 167 | $0.838 \pm 0.018$ | $0.548 \pm 0.054$ | $0.895 \pm 0.018$ | $0.758 \pm 0.071$ | $0.860 \pm 0.020$ | $0.622 \pm 0.060$ |

Table B: Editing fidelity across different levels of facial occlusion. Both CLIP-I and DINO-I decrease gradually as the occluded area increases, indicating that severe occlusions reduce the reliability of reference-guided eyeglass alignment.

## C.3 Quantitative Analysis under Facial Occlusion

We further analyze how partial occlusions influence the quality of the transferred eyeglasses. For each frame $f$, the occlusion level is estimated by a face parser and measures the proportion of pixels inside the facial region that do not belong to the detected face area or background. More formally, the occlusion ratio is defined as

$$O_f = \frac{|B_{\text{face},f} \cap M_{\text{occ},f}|}{|B_{\text{face},f}|},$$

where $B_{\text{face},f}$ denotes the convex-hull face region of frame $f$, and $M_{\text{occ},f}$ corresponds to non-face pixels located within this region. Frames are grouped into bins according to the value of $O_f$.

The results are reported in Table B using CLIP-I and DINO-I computed on the eyeglass crop. FreeEyeglass maintains consistent CLIP-I and DINO-I scores when the occluded area is small, while heavier occlusions gradually reduce both similarity measures. The same tendency is observed for MimicBrush and VACE, suggesting that partial visibility of the face primarily affects the reliability of the reference-guided alignment rather than the specific editing formulation used by our method.

## C.4 Robustness to Mask Inaccuracies

Since our method relies on a 2D reference mask to localize the eyeglasses region, we assess its sensitivity to imperfect masks. We use the same automatic mask generated by our reference-mask segmentation pipeline and evaluate three representative perturbations: (1) dilation (+2 px), (2) spatial misalignment (5-px right shift), and (3) rough hand-drawn masks. As shown in Fig. B, the edited results remain stable across all cases. The transferred eyeglasses maintain correct geometry and placement without introducing noticeable artifacts. This demonstrates that our mask-only blending strategy tolerates moderate boundary errors and does not require pixel-accurate masks.

## C.5 Robustness to Reference-Target Viewpoint Mismatch

We test our method when the reference eyeglasses are captured from a noticeably different viewpoint than the target frames (*e.g.*, left-facing reference vs. right-facing target). Despite the large pose gap, our method can still place the eyeglasses with correct orientation and produce visually coherent edits (Fig. C).

## C.6 Sequential Editing

To evaluate whether our method can compose multiple attribute edits, we apply it sequentially to different facial regions. As shown in Fig. D, we first transfer a reference nose and then apply eyeglass transfer to the edited frames. The two edits remain stable and well aligned, indicating that the method supports clean multi-attribute editing through simple sequential application.

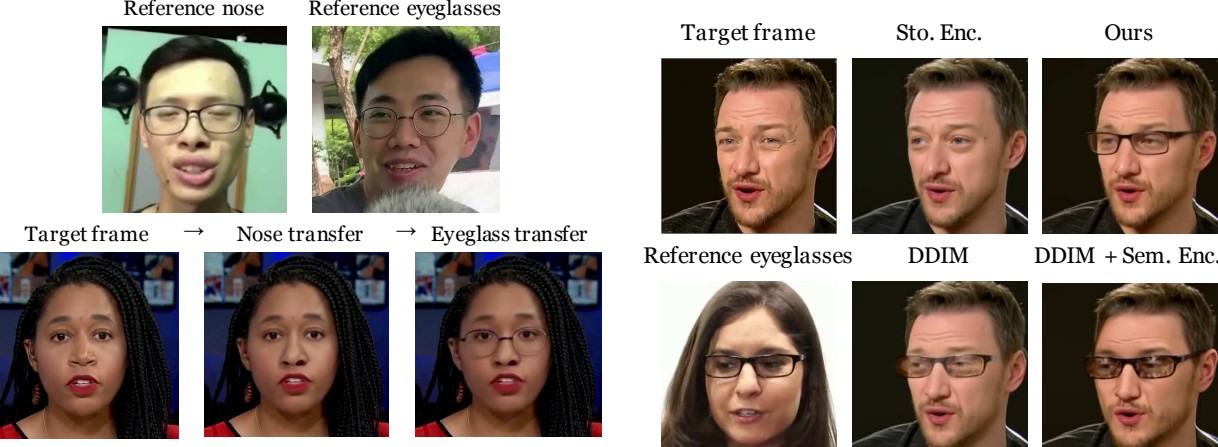

Figure D: **Sequential attribute editing.** We apply our method twice: first applying a nose transfer, followed by an eyeglass transfer to the edited result.

Figure E: **Ablations** on applying the feature blending in different components of DiffAE.

## C.7 Ablation Studies

We conduct another ablation study on the location where our feature blending strategy is applied. We apply our proposed feature-blending strategy across different components of the DiffAE: Conditional DDIM only, stochastic encoder only, conditional DDIM + semantic encoder, and semantic encoder only (Ours). Figure E shows visual results, where *Sto. Enc.* refers to the stochastic encoder. *DDIM* refers to conditional DDIM. *Sem. Enc.* refers to the semantic encoder. Applying feature blending in conditional DDIM results in unwanted artifacts and a copy-and-paste effect, *i.e.*, there are no arms for the transferred eyeglasses. Applying only to the stochastic encoder does not show successful editing results. Using feature blending in the semantic encoder achieves semantic edits.

## C.8 Analysis of Perceived Softness

Some edited frames produced by our method appear slightly softer than the target frames. We analyze the factors contributing to this effect in Fig. F.

**Approximation of stochastic details after semantic editing.** DiffAE represents images using two components: a semantic latent that controls coarse structure and a stochastic latent that captures high-frequency details. In our method, eyeglasses are transferred by blending semantic encoder features from the reference and target faces. Since the resulting semantic code does not correspond to a real edited image, the exact stochastic la-

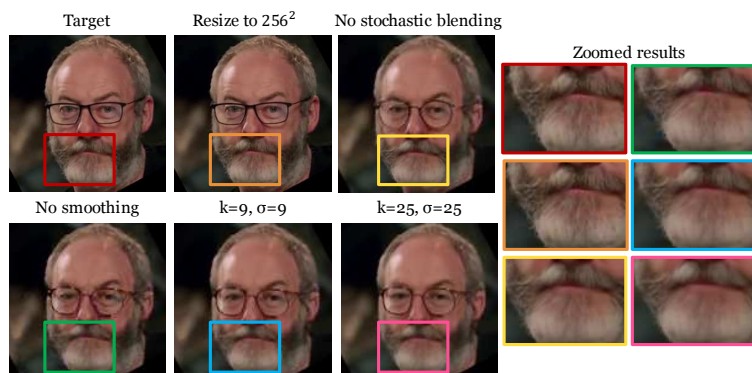

Figure F: **Analysis of factors contributing to perceived softness.** The top row examines individual components of the editing pipeline. The bottom row shows the effect of Gaussian mask smoothing during stochastic latent blending.

tent associated with this representation is unknown. We therefore approximate it by blending the inverted stochastic latents of the target and reference. While this approximation yields visually plausible edits, the stochastic latent may not perfectly match the blended semantic representation, which can slightly soften high-frequency details.

**DiffAE backbone resolution.**  Our framework uses the DiffAE-FFHQ256 model, which reconstructs images at $256 \times 256$ resolution. Aligned face regions are therefore resized to this resolution before editing.

**Gaussian smoothing for artifact suppression.**  To avoid boundary artifacts arising from mismatches between the reference and target stochastic latents, we blend them with a spatially smoothed mask. This creates a gradual transition and suppresses visible seams around the eyeglass boundary. However, enlarging the transition region mixes stochastic details from the two inputs over a wider area, which can reduce locally coherent high-frequency texture and make the result appear slightly softer.

Figure F illustrates this trade-off. Without smoothing, the edited image appears sharper, but noticeable artifacts emerge around the boundary. Increasing the smoothing strength progressively suppresses these artifacts while producing slightly softer transitions. The configuration used in the paper ($k = 25$, $\sigma = 25$) was selected to suppress prominent boundary artifacts while retaining most of the target appearance and identity.

**Potential mitigation strategies.**  Within the current framework, perceived softness can be reduced by decreasing the Gaussian smoothing strength or narrowing the softened boundary region. As shown in Fig. F, however, this recovers sharper local details at the cost of more visible seams and stochastic mismatch artifacts. The chosen configuration, therefore, reflects a practical trade-off between sharpness and boundary coherence. Beyond parameter adjustment, the effect could be further mitigated by using a higher-resolution DiffAE backbone, estimating a stochastic latent that is better matched to the blended semantic representation, or introducing a localized high-frequency refinement stage after editing. These extensions are left for future work.

### C.9   Per-video Results on the CG Dataset

To complement the averaged CG results in the main paper, we provide a per-video breakdown for all eight rendered videos in Tables C–F. We also include bar plots of PSNR, MS-SSIM, LPIPS, and MSE for the eight CG-rendered videos to improve readability in Fig. G. For clarity, we visualize a subset of representative competitive baselines rather than all methods. This analysis is included to improve transparency and clarify how performance varies across identities and between eyeglasses and sunglasses cases.

We acknowledge that the CG dataset is small, comprising four identities with one eyeglasses case and one sunglasses case each, for a total of eight videos. For this reason, we do not use the CG dataset as a standalone basis for our main claim. Instead, we use it as a controlled complementary evaluation with ground truth, alongside the larger real-video benchmark in the main paper. The per-video tables show that the overall trend reported in the averaged results remains consistent across most individual examples: our method achieves the strongest or near-strongest reconstruction fidelity on the majority of videos, particularly in PSNR and MSE, while remaining competitive in MS-SSIM and LPIPS.

### C.10   Interpreting CLIP-I and DINO-I Scores

CLIP-I and DINO-I measure visual similarity between the edited eyeglasses and the reference image, and therefore primarily capture how closely the transferred object resembles the reference appearance. However, these metrics do not explicitly evaluate whether the edited eyeglasses integrate naturally with the target face or whether identity-related facial cues are preserved.

Figure H illustrates this behavior. In this example, MimicBrush obtains higher reference similarity scores (CLIP-I: 0.944, DINO-I: 0.884) than our method (CLIP-I: 0.909, DINO-I: 0.830). A closer inspection shows that MimicBrush largely reproduces the reference eyeglass region together with parts of the surrounding eye area. Because CLIP-I and DINO-I are computed on the eyeglass crop, such direct reproduction naturally yields higher similarity scores.

In contrast, FreeEyeglass adapts the reference eyeglasses to the target face while preserving the original eye appearance and facial structure. This harmonization introduces small geometric and illumination adjustments, which can slightly reduce reference similarity scores even though the edited result better maintains the identity and motion of the target subject.

Table C: Per-video CG dataset results for PSNR (↑). Rows correspond to the eight rendered videos, and columns correspond to methods.

| Video | DVAE | VEditGAN | FLATTEN | RAVE | VidToMe | FRESCO | RF-Solver | VACE | ObjStitch | TF-ICON | PBE | AnyDoor | Mimic | OmniTry | Ours |
|---|---|---|---|---|---|---|---|---|---|---|---|---|---|---|---|
| man01–Glasses | 26.822 | 25.765 | 17.646 | 17.549 | 19.481 | 19.221 | 20.666 | 27.466 | 27.726 | 11.693 | 27.358 | 24.041 | 26.606 | 24.851 | **29.562** |
| man01–Sunglasses | 24.345 | 24.135 | 16.255 | 13.892 | 17.528 | 18.196 | 19.842 | **26.361** | 23.977 | 11.690 | 25.071 | 22.978 | 25.154 | 22.849 | 25.911 |
| man02–Glasses | 26.581 | 22.420 | 16.699 | 13.766 | 16.000 | 18.683 | 17.042 | 26.364 | 25.781 | 11.159 | 26.897 | 26.264 | 27.140 | 25.248 | **28.265** |
| man02–Sunglasses | 25.524 | 22.752 | 16.065 | 10.153 | 15.619 | 18.388 | 16.954 | 23.980 | 25.794 | 11.192 | 26.304 | 24.360 | 25.058 | 24.497 | **26.616** |
| woman01–Glasses | 27.662 | 24.229 | 13.382 | 18.594 | 17.859 | 18.182 | 19.094 | 26.598 | 26.598 | 9.184 | 28.116 | 26.677 | 27.817 | 25.189 | **30.583** |
| woman01–Sunglasses | 23.893 | 21.915 | 13.459 | 18.552 | 17.700 | 17.287 | 18.116 | 24.590 | 24.188 | 9.314 | 25.386 | 21.714 | 24.172 | 24.151 | **26.603** |
| woman02–Glasses | 24.772 | 24.760 | 13.841 | 19.304 | 18.645 | 19.235 | 18.242 | 27.829 | 28.025 | 10.254 | 27.896 | 26.165 | 26.718 | 26.478 | **29.230** |
| woman02–Sunglasses | 25.410 | 26.091 | 14.908 | 15.732 | 19.585 | 19.103 | 18.429 | 30.318 | 29.868 | 10.284 | 24.831 | 23.039 | 29.546 | 28.751 | **30.632** |

Table D: Per-video CG dataset results for MS-SSIM (↑). Rows correspond to the eight rendered videos, and columns correspond to methods.

| Video | DVAE | VEditGAN | FLATTEN | RAVE | VidToMe | FRESCO | RF-Solver | VACE | ObjStitch | TF-ICON | PBE | AnyDoor | Mimic | OmniTry | Ours |
|---|---|---|---|---|---|---|---|---|---|---|---|---|---|---|---|
| man01–Glasses | 0.947 | 0.931 | 0.731 | 0.796 | 0.861 | 0.786 | 0.527 | 0.960 | 0.963 | 0.291 | 0.960 | 0.942 | 0.965 | 0.940 | **0.973** |
| man01–Sunglasses | 0.930 | 0.923 | 0.695 | 0.765 | 0.795 | 0.766 | 0.511 | **0.955** | 0.937 | 0.311 | 0.943 | 0.932 | 0.955 | 0.924 | 0.951 |
| man02–Glasses | 0.945 | 0.881 | 0.674 | 0.801 | 0.773 | 0.787 | 0.585 | 0.955 | 0.954 | 0.339 | 0.958 | 0.960 | **0.966** | 0.947 | 0.964 |
| man02–Sunglasses | 0.940 | 0.885 | 0.664 | 0.687 | 0.750 | 0.782 | 0.581 | 0.940 | 0.947 | 0.335 | 0.950 | 0.942 | 0.953 | 0.939 | **0.957** |
| woman01–Glasses | 0.960 | 0.911 | 0.652 | 0.834 | 0.835 | 0.808 | 0.661 | 0.966 | 0.965 | 0.055 | 0.968 | 0.962 | 0.971 | 0.949 | **0.979** |
| woman01–Sunglasses | 0.938 | 0.891 | 0.640 | 0.830 | 0.826 | 0.788 | 0.642 | 0.953 | 0.948 | 0.060 | 0.955 | 0.933 | 0.955 | 0.942 | **0.963** |
| woman02–Glasses | 0.937 | 0.934 | 0.680 | 0.845 | 0.816 | 0.835 | 0.623 | 0.966 | 0.965 | 0.121 | 0.966 | 0.961 | 0.969 | 0.953 | **0.972** |
| woman02–Sunglasses | 0.941 | 0.943 | 0.682 | 0.832 | 0.850 | 0.835 | 0.627 | 0.972 | 0.971 | 0.084 | 0.952 | 0.944 | **0.976** | 0.962 | 0.974 |

Table E: Per-video CG dataset results for LPIPS (↓). Rows correspond to the eight rendered videos, and columns correspond to methods.

| Video | DVAE | VEditGAN | FLATTEN | RAVE | VidToMe | FRESCO | RF-Solver | VACE | ObjStitch | TF-ICON | PBE | AnyDoor | Mimic | OmniTry | Ours |
|---|---|---|---|---|---|---|---|---|---|---|---|---|---|---|---|
| man01–Glasses | 0.120 | 0.061 | 0.205 | 0.155 | 0.124 | 0.146 | 0.482 | 0.039 | 0.038 | 0.487 | 0.040 | 0.049 | **0.029** | 0.045 | 0.036 |
| man01–Sunglasses | 0.125 | 0.061 | 0.221 | 0.193 | 0.204 | 0.149 | 0.487 | 0.041 | 0.051 | 0.482 | 0.044 | 0.049 | **0.033** | 0.047 | 0.042 |
| man02–Glasses | 0.117 | 0.082 | 0.251 | 0.165 | 0.232 | 0.140 | 0.430 | 0.034 | 0.035 | 0.490 | 0.036 | 0.037 | **0.028** | 0.036 | 0.045 |
| man02–Sunglasses | 0.117 | 0.082 | 0.262 | 0.267 | 0.243 | 0.141 | 0.427 | 0.039 | 0.040 | 0.489 | 0.041 | 0.045 | **0.031** | 0.041 | 0.040 |
| woman01–Glasses | 0.113 | 0.083 | 0.329 | 0.164 | 0.177 | 0.174 | 0.388 | 0.039 | 0.045 | 0.552 | 0.042 | 0.045 | **0.031** | 0.040 | 0.040 |
| woman01–Sunglasses | 0.119 | 0.089 | 0.319 | 0.161 | 0.178 | 0.178 | 0.392 | 0.048 | 0.050 | 0.553 | 0.050 | 0.054 | **0.036** | 0.039 | 0.041 |
| woman02–Glasses | 0.115 | 0.059 | 0.297 | 0.137 | 0.223 | 0.128 | 0.423 | 0.032 | 0.032 | 0.483 | 0.032 | 0.032 | **0.027** | 0.036 | 0.034 |
| woman02–Sunglasses | 0.115 | 0.057 | 0.287 | 0.158 | 0.161 | 0.126 | 0.421 | 0.028 | 0.028 | 0.486 | 0.029 | 0.035 | **0.025** | 0.033 | 0.031 |

Table F: Per-video CG dataset results for MSE (↓). Rows correspond to the eight rendered videos, and columns correspond to methods.

| Video | DVAE | VEditGAN | FLATTEN | RAVE | VidToMe | FRESCO | RF-Solver | VACE | ObjStitch | TF-ICON | PBE | AnyDoor | Mimic | OmniTry | Ours |
|---|---|---|---|---|---|---|---|---|---|---|---|---|---|---|---|
| man01–Glasses | 433.29 | 533.27 | 3446.5 | 3483.1 | 2200.8 | 2346.8 | 1675.1 | 373.16 | 336.18 | 13342 | 362.34 | 260.25 | 430.19 | 649.15 | **218.29** |
| man01–Sunglasses | 725.54 | 760.19 | 4637.7 | 7968.3 | 3445.0 | 2969.7 | 2024.5 | **462.07** | 783.28 | 13350 | 619.77 | 1009.4 | 599.91 | 1021.4 | 503.212 |
| man02–Glasses | 434.00 | 1157.2 | 4275.9 | 8212.8 | 4907.9 | 2667.8 | 4065.6 | 455.63 | 522.37 | 15279 | 402.79 | 465.25 | 378.79 | 586.75 | **295.05** |
| man02–Sunglasses | 547.41 | 1073.6 | 4885.9 | 18894 | 5355.1 | 2851.3 | 4142.2 | 788.27 | 514.86 | 15202 | 457.21 | 720.54 | 612.91 | 698.02 | **433.67** |
| woman01–Glasses | 341.46 | 754.88 | 9790.9 | 2701.9 | 3220.2 | 2969.4 | 2478.5 | 428.57 | 432.29 | 23607 | 302.25 | 425.55 | 325.06 | 604.93 | **170.86** |
| woman01–Sunglasses | 804.81 | 1262.0 | 9509.9 | 2726.2 | 3330.4 | 3647.8 | 3075.8 | 685.55 | 744.97 | 22906 | 570.35 | 1320.6 | 748.49 | 771.35 | **427.98** |
| woman02–Glasses | 681.72 | 689.76 | 8486.6 | 2319.9 | 2675.5 | 2355.4 | 3066.9 | 332.44 | 314.94 | 18685 | 326.51 | 480.65 | 427.90 | 445.44 | **239.08** |
| woman02–Sunglasses | 609.06 | 515.13 | 6771.2 | 5215.2 | 2152.9 | 2426.2 | 2959.5 | 182.51 | 201.50 | 18563 | 643.45 | 996.59 | 219.23 | 261.58 | **169.540** |

(a) PSNR per video (↑)

(b) MS-SSIM per video (↑)

(c) LPIPS per video (↓)

(d) MSE per video (↓)

Figure G: Per-video quantitative results on the CG dataset. The plots visualize the values reported in Tables C–F and are provided for readability. Due to the small number of videos, they are not intended as standalone statistical evidence.

More broadly, these observations highlight that reference-similarity metrics capture only one aspect of editing quality. To obtain a more comprehensive evaluation, the main paper therefore reports multiple complementary metrics, including reference similarity (CLIP-I and DINO-I), identity preservation (ArcFace-based similarity), video fidelity (FVD and $FID_{CLIP}$), and temporal consistency measures. Together, these metrics provide a more complete assessment of both reference fidelity and target preservation in video-based eyeglass transfer.

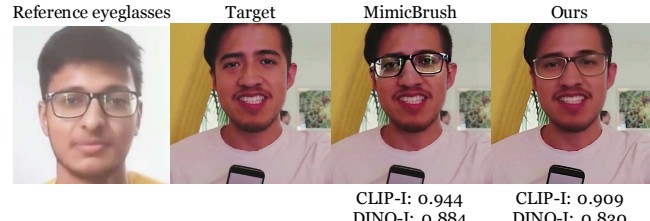

Figure H: **Example illustrating reference-similarity metrics.** MimicBrush achieves higher CLIP-I and DINO-I scores by closely reproducing the reference eyeglass region, while FreeEyeglass adapts the eyeglasses to the target face and preserves the original eye appearance.

# D  User Study

We conduct a user study to complement automatic metrics, which cannot fully capture human preferences. We ask 20 participants to rank our edited videos and the baseline results using three representative baselines: MimicBrush, VideoEditGAN, and RAVE. We randomly pick 10 videos from our evaluation benchmark and

Table G: Criteria used in our user study.

| Criterion | Question |
|---|---|
| Identity Preservation | Rank the videos in order from the one that looks most like the same person as the ID image. |
| Eyeglass Preservation | Rank the videos in order from the one that looks most like the same eyeglasses compared to the reference eyeglasses image. |
| Eyeglass Semantic Fidelity | Rank the videos in order from the one where the eyeglasses appear most naturally worn (physically natural). |
| Temporal Consistency | Rank the videos in order from the one with the most temporal consistency (least flickering). |
| Overall Video Fidelity | Rank the videos in order from the one that looks most like a real video captured by a camera. |

ask our participants to rank according to five criteria (1) Identity Preservation, (2) Eyeglass Preservation, (3) Eyeglass Semantic Fidelity, *i.e.*, is the eyeglass transfer semantically correct, (4) Temporal Consistency, and (5) Overall Video Fidelity. We provide the details of the criteria used in our user study in Table G. Table H reports the results of our user study. Our method consistently achieves the best or second-best rankings in all criteria. MimicBrush performs best in eyeglass preservation but falls short in other criteria, such as eyeglass semantic fidelity and overall video fidelity. VideoEditGAN achieves satisfying results in eyeglass semantic fidelity but ranks the lowest in eyeglass preservation. These findings highlight the strengths of our approach: while baseline methods exhibit significant weaknesses in specific criteria, our method achieves robust performance across all metrics, making it the most well-balanced solution for eyeglass transfer in facial videos.

## E Computational Analysis

We include a detailed computational time analysis for a 120-frame video on an NVIDIA A100 GPU in Table I. Our results show that the inference time of our method is comparable to that of the baseline methods, with no significant additional overhead, as it imposes no extra computational cost on the diffusion generative process.

## F Text Prompt Preparation

We use GPT 4o (OpenAI, 2024), API version 2024-05-13, to generate fine-grained descriptions of reference eyeglasses for text-based editing. We generate a fine-grained description for each reference eyeglasses and the target video, respectively. This section provides the details of generating text descriptions. We generate a fine-grained description for each reference eyeglasses and the target video, respectively. We specify our instructions with system prompts as listed in Table J. For the descriptions of target videos, we use the first frame of each video during generation. We prepare the input prompts by concatenating the descriptions of the target videos with those of the reference eyeglasses. We also impose character limits: a maximum of 100 characters for target video descriptions and 150 for reference eyeglass descriptions. To encourage photorealistic outputs, we prefix the concatenated prompts with the term *Realistic* (except for the Diffusion Video Autoencoder (Kim et al., 2023)). These prompts are then used as input for all text-based editing baselines. For FLATTEN (Cong et al., 2024), following their default configurations, we also use a negative prompt, "A person not wearing eyeglasses, deformed." Despite augmenting the prompt with the word *Realistic* and employing a negative prompt, we observe that FLATTEN frequently produced videos with a portrait drawing style rather than a photorealistic style. We suspect this tendency is due to FLATTEN's use of the Stable Diffusion 2.1 model, whereas other methods, RAVE (Kara et al., 2024) and VidToMe (Li et al., 2024), utilize the widely adopted Stable Diffusion 1.5 model.

Table H: **User study** results. We show the number of first-place rankings, mode, median, and average rank for each criterion. *Sem.* stands for semantic. **Bold** represents the best, and underline represents the second-best results.

| | Proportion of First-Place Rankings ↑ | | | |
|---|---|---|---|---|
| Method | MimicBrush Chen et al. (2024a) | VideoEditGAN Xu et al. (2022) | RAVE Kara et al. (2024) | Ours |
| ID Preservation | 0.205 | 0.300 | 0.145 | **0.350** |
| Eyeglasses Preservation | **0.370** | 0.170 | 0.215 | 0.245 |
| Eyeglasses Sem. Fidelity | 0.155 | **0.355** | 0.210 | 0.280 |
| Temporal Consistency | 0.185 | 0.320 | 0.140 | **0.355** |
| Overall Video Fidelity | 0.145 | 0.340 | 0.135 | **0.380** |

| | Mode Rank ↓ | | | |
|---|---|---|---|---|
| Method | MimicBrush Chen et al. (2024a) | VideoEditGAN Xu et al. (2022) | RAVE Kara et al. (2024) | Ours |
| ID Preservation | 3 | **1** | 4 | **1** |
| Eyeglasses Preservation | **1** | 4 | 4 | 3 |
| Eyeglasses Sem. Fidelity | 4 | **1** | 3 | **1** |
| Temporal Consistency | 3 | 2 | 4 | **1** |
| Overall Video Fidelity | 4 | **1** | 4 | **1** |

| | Median Rank ↓ | | | |
|---|---|---|---|---|
| Method | MimicBrush Chen et al. (2024a) | VideoEditGAN Xu et al. (2022) | RAVE Kara et al. (2024) | Ours |
| ID Preservation | 3 | **2** | 3 | **2** |
| Eyeglasses Preservation | 3 | 3 | **2** | **2** |
| Eyeglasses Sem. Fidelity | 3 | **2** | 3 | **2** |
| Temporal Consistency | 3 | **2** | 3 | **2** |
| Overall Video Fidelity | 3 | **2** | 3 | **2** |

| | Average Rank ↓ | | | |
|---|---|---|---|---|
| Method | MimicBrush Chen et al. (2024a) | VideoEditGAN Xu et al. (2022) | RAVE Kara et al. (2024) | Ours |
| ID Preservation | 2.585 | 2.310 | 2.890 | **2.215** |
| Eyeglasses Preservation | **2.170** | 2.780 | 2.640 | 2.410 |
| Eyeglasses Sem. Fidelity | 2.930 | **2.220** | 2.490 | 2.360 |
| Temporal Consistency | 2.655 | 2.205 | 2.970 | **2.170** |
| Overall Video Fidelity | 2.920 | 2.150 | 2.820 | **2.110** |

Table I: **Computational time analysis**. We compare the inference time taken to edit a 120-frame video on an NVIDIA A100 80GB GPU. Time is measured in seconds.

| Method | Time (s) | Method | Time (s) |
|---|---|---|---|
| ObjectStitch (Song et al., 2023) | 657 | DVAE Classifier (Kim et al., 2023) | 721 |
| TF-ICON (Lu et al., 2023) | 7505 | DVAE Text (Kim et al., 2023) | 7798 |
| Paint-by-Example (Yang et al., 2023a) | 368 | VideoEditGAN (Xu et al., 2022) | 3846 |
| Anydoor (Chen et al., 2024b) | 826 | FLATTEN (Cong et al., 2024) | 2603 |
| MimicBrush (Chen et al., 2024a) | 583 | RAVE (Kara et al., 2024) | 1157 |
| OmniTry (Feng et al., 2025) | 7444 | VidToME (Li et al., 2024) | 633 |
| VACE (Jiang et al., 2025) | 412 | RF-Solver-Edit (Wang et al., 2025) | 899 |
| **FreeEyeglass (Ours)** | 392 | FRESCO (Yang et al., 2024) | 1063 |

Table J: System prompts and user prompts used for generating fine-grained descriptions.

| System Prompt for **Eyeglasses** Descriptions |
|---|
| Task description: |
| You will receive a facial image that is wearing a pair of eyeglasses. |
| You must give a detailed description of the eyeglasses. |
| Instructions: |
| Step 1. Look at the image and locate where is the eyeglasses. |
| Step 2. State the color of the frame and also the color of the lens. |
| Determine the shape, type, and material of the frame. |
| If the frame is half-rim, state whether the top or bottom is rimless. |
| If the frame has a full rim, state whether the frame is thin or thick. |
| If there are any distinctive features of the eyeglasses, mention it also, if no just skip it. |
| Step 3. Output into a text description that only describes the eyeglasses you see. |
| Tips: The eyeglasses may be an eyeglasses or a sunglasses. |
| The shape of the eyeglasses frame includes square, boston, wellington, round, oval, aviator, browline, fox, crown panto, and geometric. |
| The fox frame is equivalent to the cateye frame. |
| The type of eyeglasses frame describes how the frame covers the lens and can be categorized into the full rim, half rim, rimless, and under-rim. |
| Output: Do not output any description that is not about the eyeglasses in the image. The final output should only focus on the eyeglasses. |

| User Prompt for **Eyeglasses** Descriptions |
|---|
| Here is the image. |

| System Prompt for **Target Video** Descriptions |
|---|
| Task description: |
| You will receive a facial image. You must give a detailed description of the image. |

| User Prompt for **Target Video** Descriptions |
|---|
| Here is the image. |

## G    Limitations and Future Work

We illustrate common failure modes of our method in Fig. I and discuss its limitations.

**Incomplete replacement due to residual target features.** In the swapping setting, failures arise when the target frame contains thick or visually dominant original glasses. Because our method edits via semantic feature blending, the target features may not be fully overwritten, leaving residual structures from the original glasses. As a result, while the color or appearance of the reference eyeglasses is transferred, the original shape may partially persist (see top row). A similar effect is observed in the removal setting, where subtle components, such as the bridge or arms, may remain (see the bottom row).

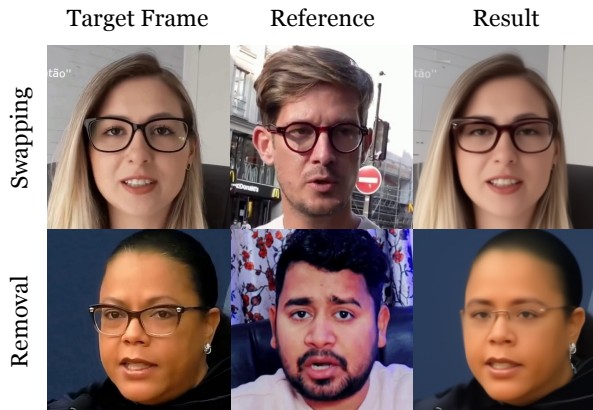

Figure I: Failure cases of our method.

**Limited geometric adaptation under large discrepancies.** As observed in our robustness analysis, performance degrades when the geometric mismatch between the reference and target is significant (e.g., large pose differences or mismatched facial proportions). Although the reconstruction prior of DiffAE enables implicit adaptation in many cases, the method does not explicitly model 3D geometry. Consequently, the transferred eyeglasses may not fully align with the underlying facial structure, leading to imperfect placement or distortion in extreme cases.

**Lighting inconsistency in lens.** Our method transfers the visual appearance of eyeglasses from the reference without explicitly modeling scene-dependent illumination or reflections. As a result, the synthesized lenses may not be fully consistent with the target video's lighting conditions. This limitation is inherent to appearance-based editing methods, including all evaluated baselines, which similarly inherit reflections from the reference rather than re-rendering them under the target illumination. Addressing this would require explicit lighting estimation or physically-based reflection modeling, which is a different problem setting and lies outside the scope of our training-free reconstruction-based framework (see Fig. 3).

These limitations arise from distinct factors, feature-level competition, geometric mismatch, and the absence of physical modeling, and point to promising directions for future work, including geometry-aware deformation and reflection-aware appearance modeling.

