# OpenReview forum: "FreeEyeglass: Training-free and Target-mask-free Eyeglass Transfer for Facial Videos"
_TMLR — Decision pending for TMLR_

### Review · Reviewer_wvEM · 2026-02-24

**Summary Of Contributions:**

The paper presents FreeEyeglass, a training-free framework for transferring eyeglasses onto target facial videos. The method utilizes a Diffusion Autoencoder. The core mechanism involves extracting semantic and stochastic latent codes from both the target video frames and a single reference eyeglass image. The authors introduce a feature blending operation within the semantic encoder, merging reference and target features using a 2D binary mask derived from the reference image. To maintain temporal consistency across video frames, the method replaces standard spatial self-attention with regional optical-flow-guided self-attention confined to the predefined region of interest of the eyeglasses.

**Audience:**

Yes

**Audience Explanation:**

The problem studied in this paper is of great practical value, especially in the ecommercial area. The framework does not require frame-by-frame mask annotations for the target video or additional model fine-tuning. It operates entirely using inference-time latent manipulation.

**Claims And Evidence:**

No

**Claims Explanation:**

1. The blending is executed via the equation $R_{i}(x)=M\odot R_{i}(x_{ref})+(1-M)\odot R_{i}(x)$, where $M$ is the mask derived from the reference image. This operation performs a rigid spatial replacement in the feature domain. Because $M$ is strictly tied to the spatial coordinates of the reference face, applying it directly to the target features assumes near-perfect geometric alignment between the two faces. While the faces are pre-aligned to a standard crop, exact facial proportions (e.g., interpupillary distance) and 3D head poses will inevitably differ. The authors acknowledge this limitation, noting that the method fails when head yaw angles exceed $\pm40-50^{\circ}$ due to "strong geometric mismatch". The linear spatial blending fails to deform or adapt the reference features to the target's underlying 3D geometry.

2. The method merely transfers the lens appearance directly from the reference image. The authors concede that achieving physically consistent reflections would require environment mapping or illumination modeling, which is not supported by their training-free reconstruction architecture. Consequently, the eyeglasses will retain the lighting environment of the reference image, leading to a physical discrepancy when inserted into a target video with different ambient lighting.

3. The benchmark utilizes a newly constructed dataset from CelebV-HQ containing 100 video pairs. Because this real-world data lacks ground truth, the authors supplement it with a CG-rendered dataset. However, this CG dataset only contains 4 identities and 2 pairs of glasses, totaling just 8 ground-truth videos. Evaluating standard quantitative metrics (PSNR, SSIM, MSE) on a sample size of $N=8$ provides limited statistical significance.

4. The authors admit the method fails under specific conditions. However, these limitations are only demonstrated qualitatively. The paper lacks a quantitative breakdown of how the metrics degrade as pose angles increase or as occlusion percentages rise

**Requested Changes:**

Please refer to the above comments.

---

> ### Author Response · Authors · 2026-03-17
>
> We thank the reviewer for the thoughtful and detailed feedback and provide clarifications below to address several points regarding geometric alignment, physical consistency, and the evaluation design.
>
> **1. On the concern that feature blending assumes perfect geometric alignment**
>
> The reviewer notes that Eq. (1) performs spatial blending using a reference mask and suggests that this operation assumes near-perfect geometric alignment between the reference and target faces. We would like to clarify that the blending is not performed in pixel space but in the semantic feature space of DiffAE’s encoder. These semantic features encode high-level facial structure rather than raw spatial appearance, and the blended representation is subsequently interpreted by the decoder together with the target frame’s stochastic latent during reconstruction. As a result, the transferred eyeglasses are synthesized in a manner consistent with the target face geometry rather than being rigidly copied from the reference.
>
> To further examine this behavior, we conducted a quantitative pose analysis by grouping frames according to head yaw angle.
> Table R1 reports CLIP-I scores across yaw bins for our method, and we additionally evaluate strong baselines (MimicBrush and VACE) under the same protocol. For FreeEyeglass, editing fidelity remains stable across pose ranges, with CLIP-I varying only slightly from 0.870 (<10°) to 0.851 (>50°). Importantly, we observe similar degradation trends for both MimicBrush and VACE as the pose becomes more extreme. This indicates that the performance drop at very large viewpoints reflects the inherent difficulty of reference-guided eyeglass transfer under large pose differences, rather than a limitation of our feature-blending formulation.
>
> **Table R1: CLIP-I Performance Across Yaw Angles**
>
> | Yaw Range | # Frames | Ours ↑ | MimicBrush ↑ | VACE ↑ |
> |:----------|---------:|-------:|-------------:|-------:|
> | < 10°     | 4,397    | 0.870  | 0.917    | 0.869  |
> | 10–20°    | 3,300    | 0.868  | 0.906    | 0.852  |
> | 20–30°    | 1,976    | 0.864  | 0.909    | 0.853  |
> | 30–40°    |   980    | 0.854  | 0.898    | 0.864  |
> | 40–50°    |   975    | 0.850  | 0.899    | 0.845  |
> | > 50°     |   272    | 0.851  | 0.885    | 0.830  |
>
> Overall, both the formulation and empirical evidence show that our method does not rely on near-perfect geometric alignment, but instead leverages DiffAE’s reconstruction prior to adapting the transferred eyeglasses to the target face.
>
>
> **2. On the lighting consistency of eyeglass reflections**
>
> Our framework focuses on appearance transfer under a reconstruction-based editing paradigm rather than physically-based rendering. As such, it does not explicitly model environment lighting or reflection synthesis.
> Importantly, none of the evaluated baselines perform reflection-aware rendering either. Instead, all methods, including diffusion-based inpainting approaches, transfer the visual appearance of the reference object.
> We therefore position our contribution as enabling semantically consistent and identity-preserving editing without target masks or training, rather than solving physically-based rendering. We will clarify this scope explicitly in the revision to avoid ambiguity.
>
> **3. On the size of the CG dataset**
>
> The reviewer raises concerns about the statistical significance of the CG evaluation given its small sample size.
> We would like to clarify that the CG dataset is not our primary benchmark. The main evaluation is conducted on a 100-pair real-video benchmark constructed from CelebV-HQ, which forms the basis for the quantitative results in Table 2 of our paper.
> The CG dataset serves only as a supplementary evaluation with ground-truth frames, enabling the computation of pixel-level metrics that cannot be obtained from real videos.
>
> Despite its small size (8 videos), the results are highly consistent across metrics:
>
> * PSNR / MSE: best on 7/8 videos, second-best on the remaining one
> * MS-SSIM: best on 5/8, second-best on 2/8
> * LPIPS: consistently competitive, including second-best performance
>
> This level of consistency indicates that the observed improvements are systematic rather than incidental, and align with the conclusions drawn from the main benchmark. Expanding the CG dataset is constrained by the need for professionally created assets and manual scene design, which makes large-scale generation difficult within the revision period.
> To improve clarity and transparency, we will (i) explicitly clarify the role of the CG dataset in the main paper, and (ii) provide full per-video results for all metrics in the supplementary material.

---

> > ### Author Response · Authors · 2026-03-17
> >
> > **4. On the lack of quantitative analysis for failure conditions**
> >
> > We thank the reviewer for raising concerns about robustness to large pose variations and occlusions. The pose and occlusion experiments in Sec. 4.4 were intended to illustrate the operational range of the method rather than to indicate a fundamental weakness. Following the reviewer’s suggestion, we now include a quantitative breakdown across pose and occlusion bins, which strengthens the robustness analysis in the paper. We will include it in the main paper.
> >
> > **Pose variation.**
> > As shown in Table R1, the pose-binned analysis shows that editing fidelity remains stable across moderate head rotations, with CLIP-I decreasing only slightly from 0.870 at near-frontal views to 0.850–0.851 at large yaw angles. The baselines exhibit similar trends, suggesting that the degradation at extreme poses primarily reflects the intrinsic difficulty of the task rather than a methodological limitation of FreeEyeglass.
> >
> > **Occlusion.**
> > We also analyze the effect of facial occlusion by estimating, for each frame, the fraction of non-face pixels within the convex-hull face region and grouping frames into occlusion bins. The results show that editing fidelity remains stable under typical occlusion levels (≤10%), where CLIP-I remains around 0.864–0.869, and decreases gradually to 0.838 under heavier occlusion (≥15%). Similar trends are observed for the baseline methods. These results indicate that moderate facial occlusions have a limited impact on reference-guided eyeglass transfer, while larger occlusions naturally increase the difficulty of reconstructing the occluded regions.
> >
> > **Table R2. CLIP-I performance across occlusion levels.**
> >
> > | Occlusion Range | # Frames | Ours ↑ | MimicBrush ↑ | VACE ↑ |
> > |:----------------|---------:|-------:|-------------:|-------:|
> > | 0–5%            | 4,136    | 0.869  | 0.912    | 0.859  |
> > | 5–10%           | 7,046    | 0.864  | 0.907    | 0.860  |
> > | 10–15%          |   551    | 0.854  | 0.904    | 0.838  |
> > | ≥15%            |   167    | 0.838  | 0.895    | 0.860  |
> >
> > We believe these clarifications and additional analyses resolve the reviewer’s concerns, and the revised manuscript reflecting these updates will be uploaded shortly.

---

> ### Author Response · Authors · 2026-03-17
> **We have revised the manuscript**
>
> We thank the reviewer for the detailed and insightful comments. We have uploaded a revised manuscript addressing the concerns raised. In particular, we (i) **add a quantitative robustness analysis under pose and occlusion** (Sec. 4.4, Supp. Sec. C.2–C.3), (ii) **clarify the feature blending mechanism** to emphasize its operation in semantic latent space with implicit geometric adaptation (Sec. 3.2), (iii) **provide per-video CG dataset results** for transparency (Supp. Sec. C.9), and (iv) **explicitly discuss lighting limitations**  (Conclusion and Supp. Sec. G). We hope these revisions address the reviewer’s concerns.

---

### Review · Reviewer_ejj7 · 2026-03-02

**Summary Of Contributions:**

This submission proposes a training-free framework for transferring eyeglasses from a reference image to facial videos using Diffusion Autoencoders by combining feature blending in the semantic encoder with spatial-temporal self-attention to achieve local editing without requiring per-frame target masks. While the training-free and mask-free aspects are practically appealing, and the extensive experimental comparison against numerous baselines is carried, the work has significant weaknesses: the technical contribution is incremental (largely blending features and adapting existing attention mechanisms), the reliance on a reference mask still requires manual annotation or external segmentation tools, the method struggles with large pose variations and occlusions as acknowledged by the authors, and the evaluation metrics are inconsistently applied with some favoring the method's reconstruction bias over actual editing quality. The claimed advantages over inpainting-based approaches are modest.

**Additional Comments:**

N.A.

**Audience:**

Yes

**Audience Explanation:**

On one hand, the studied problem is a subtask in the AIGC community. I believe there should be some audience. On the other hand, I would not expect much attention, as this task is too specific, so it is less important compared with more-general tasks.

**Broader Impact Concerns:**

N.A.

**Claims And Evidence:**

No

**Claims Explanation:**

Besides the weakness mentioned above, in terms of the evaluation/evidence, there are drawbacks listed as follows.

1. I found that the results from the proposed method is typically blurred compared with the given input. For example, the results in Figure 3 show the pattern (last column).

2. I suppose for this task, a more important metric is the face identity fidelity. While I did not find the evaluation results in Table 3.

3. Artifacts: in Figure 3, the last row, the eyeglass has changed, with regard to the given eyeglass.

**Requested Changes:**

I would expect such revisions/explanations:

1. why result images are blur?

2. ID fidelity metric results.

---

> ### Author Response · Authors · 2026-03-17
>
> We thank the reviewer for the detailed and constructive feedback. Following the suggestions, we have strengthened the paper by (i) adding identity fidelity metrics to quantify preservation of facial identity explicitly, and (ii) clarifying the source of the slight smoothing effect observed in our results. The updated evaluation shows that our method consistently preserves identity while maintaining temporal consistency.
>
> We also clarify several points regarding the perceived weaknesses.
> First, while the individual components are conceptually simple, our contribution lies in demonstrating that reference-guided localized editing can be made reliable within the reconstruction-oriented latent space of DiffAE, without training or per-frame target masks. This setting imposes unique constraints, and naïve combinations of these components fail to achieve stable editing, as shown in Sec. 4.6.
>
> Second, regarding the use of a reference mask, we emphasize that our method requires a mask only for the reference image, not the target video. This differs from inpainting-based approaches that require per-frame target masks, which are costly to obtain in videos; in practice, the reference mask can be generated automatically using standard segmentation tools, and our experiments show the method is robust to imperfect masks.
>
> Third, the pose and occlusion analyses are intended to characterize the operational range of the method rather than indicate a fundamental limitation. In the revised manuscript (Sec. 4.4, Table 4 and 5), we now include a quantitative breakdown across pose and occlusion bins. The results show that performance degrades only gradually at extreme poses, and similar trends are observed across baselines, suggesting that this reflects intrinsic task difficulty rather than a method-specific weakness.
>
> Finally, our evaluation is designed to capture complementary aspects of video editing, including editing fidelity, temporal consistency, and target preservation. While some methods achieve higher similarity scores by closely copying reference appearance, this can come at the cost of identity preservation or temporal stability in videos. We therefore report all metrics separately and include identity fidelity in the revision, ensuring a fair and balanced comparison.
>
> We now address each of the reviewer’s specific concerns regarding blur, identity fidelity, and visual artifacts in turn.
>
> **1.  Why do the results appear blurred?**
>
> The primary reason is the approximation of the stochastic latent after semantic editing. In DiffAE, images are represented by a semantic latent (coarse structure) and a stochastic latent (high-frequency details). After blending semantic features from the reference and target faces, there is no real edited image from which to infer the exact stochastic latent. We therefore approximate it by blending the inverted stochastic latents of the target and reference, which produces plausible edits but may slightly soften high-frequency details. In addition, the DiffAE backbone operates at 256×256 resolution, and we apply a smoothed mask when blending stochastic latents to suppress boundary artifacts. As shown in the supplementary analysis, reducing the smoothing yields sharper edges but introduces visible boundary inconsistencies, while stronger smoothing produces cleaner integration with slightly softer textures.

---

> > ### Author Response · Authors · 2026-03-17
> >
> > **2.  Identity fidelity evaluation**
> >
> > We agree that identity preservation is an important metric for this task. Following the reviewer’s suggestion, we evaluate identity fidelity using ArcFace similarity between edited frames and the original target frames.
> > Some methods, such as DVAE-Classifier and VideoEditGAN, obtain higher ArcFace similarity scores; however, these approaches are not reference-based editing methods and largely preserve the original facial appearance without explicitly transferring a new eyeglass style from a reference image. As a result, their higher identity similarity partly reflects weaker appearance modification.
> > Among methods that reliably perform reference-guided eyeglass transfer, our method achieves identity similarity comparable to strong video editing baselines such as VACE while maintaining consistent eyeglass appearance across frames. In addition, our approach operates entirely at inference time without additional training or fine-tuning, which makes it practical for reference-based video editing scenarios.
> >
> > **Table R3. Quantitative comparison of identity preservation and editing fidelity.**
> > | Method                          | ArcFace ID ↑ | CLIP-I ↑  | DINO-I ↑  |
> > | ------------------------------- | ------------ | --------- | --------- |
> > | **Image-based Editing Methods** |              |           |           |
> > | Object Stitch                   | 0.593        | 0.882     | 0.624     |
> > | TF-ICON                         | 0.110        | 0.802     | 0.441     |
> > | Paint-by-Example (PBE)          | 0.568        | 0.877     | 0.650     |
> > | AnyDoor                         | 0.559        | 0.850     | 0.517     |
> > | MimicBrush                      | 0.582        | 0.909     | 0.739     |
> > | OmniTry                         | 0.511        | 0.857     | 0.588     |
> > | **Video Editing Methods**       |              |           |           |
> > | VideoEditGAN                    | 0.642        | 0.846     | 0.547     |
> > | DVAE Classifier                 | 0.732        | 0.819     | 0.433     |
> > | FLATTEN                         | 0.381        | 0.780     | 0.300     |
> > | RAVE                            | 0.150        | 0.833     | 0.526     |
> > | VidToMe                         | 0.340        | 0.828     | 0.499     |
> > | FRESCO                          | 0.142        | 0.787     | 0.486     |
> > | RF-Solver-Edit                  | 0.363        | 0.833     | 0.492     |
> > | VACE                            | 0.665        | 0.858     | 0.634     |
> > | **FreeEyeglass (Ours)**         | 0.622    | 0.865 | 0.542 |
> >
> >
> > **3. Artifact in Figure 3 (last row)**
> >
> > We appreciate the reviewer for pointing this out.
> > The example corresponds to a glasses-swapping scenario. In this case, the semantic encoder transfers the appearance of the reference glasses but sometimes partially retains geometric cues from the original glasses, leading to a mixed shape.
> > We will revise the figure to include a clearer example and discuss this limitation in the supplementary.
> >
> > The revised manuscript incorporating these changes will be uploaded shortly. We hope that the additional analyses and clarifications provide a clearer, more complete picture of the method’s strengths in achieving a balanced trade-off between identity preservation, temporal consistency, and reference-guided editing.

---

> ### Author Response · Authors · 2026-03-17
> **We have revised the manuscript**
>
> We thank the reviewer for the detailed feedback. We have uploaded a revised manuscript addressing the concerns raised. Specifically, we (i) **add ArcFace-based identity fidelity metrics** in Sec. 4.2, (ii) **clarify the source of the slight smoothing effect with additional analysis** (Sec. 4.2 and Supp. Sec. C.8), (iii) **revise the qualitative example in Fig. 3** with a more representative case and **expand the discussion of the corresponding failure case** (Supp. Sec. G), (iv) **include a quantitative robustness analysis under pose and occlusion** (Sec. 4.4), and (v) **refine the introduction** to better clarify the method’s scope and contributions (Sec. 1). We hope these revisions address the reviewer’s concerns.

---

### Review · Reviewer_2DiQ · 2026-03-03

**Summary Of Contributions:**

The paper introduces FreeEyeglass, a novel framework designed for reference-based eyeglass transfer in facial videos. Unlike existing methods that often rely on per-frame masks or extensive fine-tuning, FreeEyeglass is both training-free and target-mask-free. The authors leverage the latent space of a Diffusion Autoencoder (DiffAE) to achieve high-fidelity reconstruction and identity preservation.
Key technical contributions include:

- A feature-blending mechanism within the DiffAE semantic encoder that merges reference eyeglass features with target facial features.
- An inflated pseudo-3D U-Net architecture using regional flow-guided self-attention to ensure temporal stability across video frames.
- The establishment of a new evaluation benchmark for video eyeglass transfer based on the CelebV-HQ dataset.


Strengths
- Methodological Soundness: The decision to use Diffusion Autoencoders (DiffAE) instead of standard text-to-image diffusion models is well-justified. By focusing on the reconstruction-oriented latent space, the method maintains facial identity much more effectively than inpainting-based baselines.
- Practical Utility: The training-free nature of the model makes it highly accessible for real-world applications. Removing the need for precise target masks significantly reduces the preprocessing overhead for video editing.
- Superior Performance: Quantitative results demonstrate that FreeEyeglass achieves the best balance between editing quality and temporal consistency (S_edit score). It significantly outperforms state-of-the-art methods like AnyDoor and Shape-guided diffusion in identity preservation (TL-ID and TG-ID metrics).
- Temporal Consistency: The flow-guided attention mechanism successfully addresses the common flickering problem in video editing, leading to smooth transitions and stable eyeglass positioning.
- Supplementary Material: Video results are provided, which makes the method convincing.

Weaknesses
- Minor Metric Trade-offs: The model shows slightly lower scores in CLIP-I and DINO-I compared to some baselines. The authors argue this is because these metrics reward copy-pasting which ignores facial harmonization. While plausible, more qualitative evidence showing why the harmonized result is superior to the copy-pasted result would strengthen the argument.
- Complex Scenarios: As noted in the failure cases, the model struggles when the target already wears thick-rimmed glasses, as it sometimes merges the shapes rather than completely replacing them.
- Lighting Consistency: The framework does not currently account for target-scene lighting or environment-mapped reflections on the lenses, which can occasionally affect the realism of the composite.

**Audience:**

Yes

**Audience Explanation:**

Ys

**Claims And Evidence:**

Yes

**Claims Explanation:**

Yes

**Requested Changes:**

See above

---

> ### Author Response · Authors · 2026-03-17
>
> We thank the reviewer for the careful reading of our paper and the positive assessment of the methodological design and experimental validation. We are glad that the reviewer finds the DiffAE-based reconstruction framework well justified and recognizes the practical value of a training-free, target-mask-free approach. Below, we address the concerns regarding metric interpretation, complex scenarios, and lighting consistency.
>
> **1. Metric Trade-offs (CLIP-I and DINO-I)**
>
> Thank you for the suggestion to provide qualitative evidence clarifying the slight differences in CLIP-I and DINO-I. We agree that this helps interpret the quantitative results.
> In the revised supplementary material, we add an additional comparison (Fig. G). In this example, MimicBrush obtains higher CLIP-I and DINO-I scores (0.944 / 0.884) than our method (0.909 / 0.830). However, visual inspection shows that MimicBrush largely reproduces the reference eyeglasses region, including the eye area. Since CLIP-I and DINO-I measure similarity between the edited eyeglasses crop and the reference image, such direct reproduction naturally leads to higher scores.
> In contrast, our method adapts the reference eyeglasses to the target face while preserving the original eye appearance. This harmonization introduces small geometric and illumination adjustments, which may slightly reduce reference similarity scores despite producing results that better preserve identity and integrate more naturally with the target face.
>
> This example highlights a limitation of reference-similarity metrics: they reward direct visual similarity to the reference object, even when this comes at the cost of identity preservation and facial coherence. In contrast, the goal of eyeglass transfer is not to replicate the reference region, but to adapt it to the target face while preserving identity and maintaining natural integration.
> To better reflect this objective, we complement CLIP-I and DINO-I with eye-region preservation and temporal identity consistency metrics, which more directly measure whether the edited results remain faithful to the target subject across frames.
>
>
> **2. Complex Scenarios**
>
> We appreciate the reviewer’s observation regarding cases where the target subject already wears thick-rimmed glasses. In these cases, the model may partially merge the original glasses with the reference appearance rather than fully replacing them. When the target glasses are visually dominant, some target eyeglass information can persist during semantic feature blending. We clarify this limitation in the revised manuscript and include representative examples in the failure case section.
>
>
> **3. Lighting Consistency and Reflections**
>
> We agree that physically consistent reflections could further improve realism. Our framework focuses on appearance transfer within a reconstruction-based editing paradigm, and does not explicitly model scene lighting or reflection synthesis. Modeling such effects would require explicit estimation of scene lighting and re-rendering, which lies outside the scope of our training-free framework.
> We position our contribution as enabling semantically consistent and identity-preserving editing without target masks or training, rather than addressing physically-based rendering. We clarify this scope explicitly in the revised manuscript to avoid ambiguity.
>
> We have incorporated all of the above clarifications and additional qualitative evidence into the revised manuscript and supplementary material, and we will upload the updated version shortly.

---

> ### Author Response · Authors · 2026-03-17
> **We have revised the manuscript**
>
> We thank the reviewer for the positive and constructive feedback. We have uploaded a revised version of the manuscript addressing the points raised. In particular, we (i) **clarify the interpretation of CLIP-I/DINO-I** (Supp. Sec. C.10) , (ii) **expand the discussion of failure cases** (Supp. Sec. G), and (iii) explicitly state the **limitation regarding lighting and reflections** (Conclusion and Supp. Sec. G). We hope these revisions further strengthen the clarity and completeness of the paper.

---

### Author Response · Authors · 2026-03-17

Dear Action Editor and Reviewers,

We thank the reviewers for the detailed and constructive feedback. Following the suggestions, we have prepared additional results to strengthen the empirical support of the paper, including **identity fidelity metrics** and **quantitative analyses across pose and occlusion conditions**. We also provide a clearer **analysis of the smoothing effect**, along with **an extended evaluation on the CG dataset** and a more detailed **interpretation of CLIP-I and DINO-I**.  We will also revise the wording of several claims to improve clarity and avoid potential ambiguity. These changes will be included in the revision.

We aim to provide stronger empirical support for the paper’s main claim that **reference-guided localized editing can be reliably achieved in the reconstruction-based latent space of DiffAE**, enabling identity-preserving and temporally consistent video editing without training or per-frame target masks. We will revise the paper to make this claim and its supporting evidence clearer.

We also clarify that our method focuses on transferring visual appearance within a reconstruction-based editing framework. It does not explicitly model lighting effects, which we discuss as limitations, and instead focuses on producing stable edits that preserve identity and remain consistent over time.

We believe these additions and clarifications address the main concerns regarding experimental support and better reflect the scope and contributions of the work. We will upload the revised manuscript with these updates shortly for the reviewers’ consideration.

---

> ### Author Response · Authors · 2026-03-17
> **Summary of Changes**
>
> We thank the reviewers for their constructive feedback. We have revised the manuscript as follows:
> 1. **Identity fidelity evaluation** (Sec. 4.2, Table 3).
> We add ArcFace-based identity similarity metrics to explicitly quantify identity preservation, and revise the discussion to incorporate these results.
> 2. **Clarification of smoothing effect** (Sec. 4.2 and Supp. Sec. C.8).
> We add an explanation that the slight smoothing arises from stochastic latent approximation after semantic editing and boundary smoothing during latent blending, and include an ablation analysis demonstrating the trade-off between sharpness and boundary artifacts.
> 3. **Quantitative robustness analysis** (Sec. 4.4, Tables 4–5, Supp. Sec. C.2–C.3).
> We add a quantitative breakdown under (i) head pose (yaw bins) and (ii) occlusion levels. The results show stable performance under moderate conditions with gradual degradation only in extreme cases.
> 4. **CG dataset clarification and transparency** (Sec. 4.1, Sec. 4.3, and Supp. Sec. C.9).
> We clarify the role of the CG dataset as a controlled complementary evaluation in the dataset section and add a pointer in the results section to detailed per-video results. We further provide a full per-video breakdown (PSNR, MS-SSIM, LPIPS, MSE) for all eight sequences in the supplementary.
> 5. **Method clarification on feature blending** (Sec. 3.2).
> We revise the description around Eq. (1) to clarify that blending is performed in the semantic latent space rather than pixel space, resulting in non-rigid integration with implicit geometric adaptation instead of rigid spatial replacement.
> 6. **Lighting limitation clarification** (Conclusion and Supp. Sec. G).
> We explicitly state that the method does not model scene illumination or reflections, and clarify that this limitation is shared by all evaluated baselines. We position reflection-aware rendering as future work.
> 7. **Expanded discussion of limitations** (Supp. Sec. G).
> We extend the analysis of failure modes, including cases with dominant original glasses and large geometric mismatch, and clarify the scope and limitations of the approach.
> 8. **Analysis of CLIP-I and DINO-I behavior** (Supp. Sec. C.10).
> We add a supplementary analysis and visualization to explain the behavior of CLIP-I and DINO-I, clarifying that these metrics may favor copy-paste appearance over harmonized results.
> 9. **Clarity improvements** (Sec. 1).
> We refine the wording in the introduction to improve clarity of the method description and contributions.
> 10. **Improvement of qualitative results** (Fig. 3).
> We revise the last row of the main qualitative figure to showcase a more accurate and representative example.
>
> Overall, these revisions strengthen the evaluation, improve methodological clarity, and directly address the concerns raised by the reviewers.

---

### Author Response · Authors · 2026-05-27
**Status Inquiry**

Dear Action Editor,

We hope you are doing well. We would like to kindly inquire about the current status of our submission.

The submission status has remained "Decision Pending" for about a month since we submitted our rebuttal and revisions, so we were wondering whether there are any updates regarding the review process or expected next steps.

Thank you very much for your time and consideration.

Authors

---

### Decision · Action_Editor_SG6R · 2026-04-17

**Recommendation:** Accept with minor revision

**Additional Comments:**

One reviewer raised concerns regarding the results, particularly the issue of blur in the generated outputs. The authors should provide a convincing explanation for this observation in the revised manuscript and discuss potential approaches to mitigate or address it.

**Audience:**

Yes

**Audience Explanation:**

Yes. Although the problem setting is relatively specific, the paper addresses a practically relevant task within the broader area of generative modeling and video editing, which is of clear interest to parts of the TMLR audience. The training free and mask free formulation, along with the use of diffusion autoencoder latent manipulation, offers insights that can extend to other localized and temporally consistent editing tasks. Researchers working on controllable generation, video synthesis, and applied AIGC systems, including domains such as virtual try on and media editing, are likely to find the findings useful.

**Claims And Evidence:**

Yes

**Claims Explanation:**

Yes. The claims are largely supported by comprehensive experimental evidence, including both quantitative metrics and qualitative comparisons against strong baselines. The authors demonstrate improvements in identity preservation, editing quality, and temporal consistency, and further strengthen their case by adding ArcFace-based identity metrics and robustness analyses across pose and occlusion conditions in the revision. The inclusion of ablation studies and supplementary video results helps validate the design choices and practical effectiveness of the method. While some metrics and edge cases reveal limitations, these are transparently discussed and do not undermine the overall validity of the claims.